# MATCHED-PAIR EXPERIMENTAL DESIGN WITH ACTIVE LEARNING

## ABSTRACT

Matched-pair experimental designs aim to detect treatment effects by pairing participants and comparing within-pair outcome differences. In many situations, the overall effect size across the entire population is small. Then, the focus naturally shifts to identifying and targeting high treatment-effect regions where the intervention is most effective. This paper proposes a matched-pair experimental design that sequentially and actively enrolls patients in high treatment-effect regions. Importantly, we frame the identification of the target region as a classification problem and propose an active learning framework tailored to matched-pair designs. Our design not only reduces the experimental cost of detecting treatment efficacy, but also ensures that the identified regions enclose the entire high-treatment-effect regions. Our theoretical analysis of the framework's label complexity and experiments in practical scenarios demonstrate the efficiency and advantages of the approach.

## 1 INTRODUCTION

Matched-pair experimental designs (MPED) group participants with similar properties into pairs, randomly assigning the treatment to one participant in each pair and the control to the other. This design enables experimenters to compare the treatment and control outcomes within pairs, reducing the variance in the difference between treatment and control outcomes to determine the effectiveness of the treatment. Hence, MPED is a conventional technique used in causal inference to draw valid conclusions for an intervention using a limited sample size (Stuart, 2010). For instance, policy-

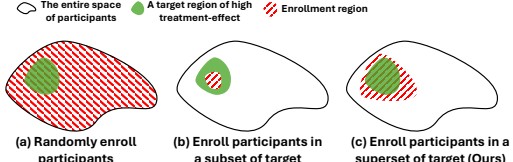

(a) Randomly enroll participants    (b) Enroll participants in a subset of target    (c) Enroll participants in a superset of target (Ours)

Figure 1: The enrollment region of our active design **(c)** encloses the target region with a high treatment effect. The conventional MPED **(a)** mainly enrolls unresponsive participants, leading to inefficiency. Existing active designs **(b)** risk focusing on a subset of the target, missing many true responders.

makers, clinicians, and web developers conduct MPED to evaluate the impact of a new policy, drug, or website design. More details for MPED can be found in Goswami et al. (2015); Welsh et al. (2023).

When the treatment effect across the entire population is small, MPED may lack power with small sample sizes. In this work, we tackle the problem of *detecting* treatment efficacy in MPED under the constraint of the experimental budget that only a limited number of patients are permitted to receive experimental interventions (or treatments).

**Related Work**    Studies such as Simon & Simon (2013); Burnett & Jennison (2021); Thall (2021) emphasize the practical need to enroll participants who are highly responsive to the treatment when the effect size in the entire population is small. To address this challenge, these authors developed methodologies to actively select participants from sub-populations with high treatment effects, thereby enabling experimenters to efficiently identify responder regions. However, their designs are motivated by randomized controlled trials (RCTs), i.e., randomly assigning treatment and control to patient units without pairing patients. Another line of relevant research focuses on the estimation of the Conditional Average Treatment Effect (CATE). For example, works such as Jesson et al. (2021); Piskorz et al. (2025); Shalit et al. (2017); Alaa & Van Der Schaar (2017; 2018) propose actively enrolling patients into experiments to efficiently estimate individual treatment effects. These studies

are fundamentally estimation problems, where the objective is to quantify the treatment effect size with a limited sample. In contrast, our work is developed with the goal of *detecting* the existence of a treatment effect. Moreover, our proposed method is a *sequential design* that actively enrolls patient responders and sequentially evaluates the existence of the treatment effect through modeling aimed at identifying the responders.

**In this paper**, we propose an active-learning-based design tailored to MPED to enroll participants from high treatment-effect regions, addressing scenarios where the average treatment-effect size is small across the entire population. We reformulate the identification of high treatment-effect regions as a classification problem and employ active learning (Hanneke et al., 2014; Balcan et al., 2006) to address it under a limited experimental (or label) budget. This reformulation as a classification problem is exclusive to MPED, offering the distinct benefit of "*enclosing target regions*". As illustrated in Figure 1, our design enrolls participants from a superset of the high treatment-effect target, enhancing experimental efficiency while ensuring that, when a treatment is deemed effective, its enrollment region includes the target population of responders. *This has high clinical value: Many existing active experimental designs produce enrollment regions with insufficiently revealed responders, which can lead to the false conclusion that a treatment is not broadly applicable and cause the premature termination of a study. In this paper, we present an active and sequential design with theoretical guarantees and practical value to mitigate this issue.*

Our contributions are summarized as follows:

- We develop an active-learning-based design, termed *MPED-RobustCAL*, for MPED. This design is *active and sequential*: it actively learns a classifier to identify regions with high treatment effects, while sequentially enrolling participants from these regions to test whether a treatment is effective.
- We conduct a theoretical analysis of *MPED-RobustCAL*, demonstrating that the enrollment region encloses and converges to the target region more efficiently compared to passive learning.
- We present a practical instantiation of *MPED-RobustCAL* and evaluate it through simulations on synthetic data as well as two real datasets. The results demonstrate the advantages of *MPED-RobustCAL* over conventional approaches, providing empirical support for our theoretical analysis.

## 2 PRELIMINARIES

In this section, we present the preliminaries of MPED, including the data model for generating experimental data, the conventional MPED, and the two-sample testing problem.

### 2.1 DATA MODEL

Let $p_{\mathbf{X}}(\mathbf{x})$ denote the probability density function (pdf) from which a participant, represented by covariates $\mathbf{X} \in \mathbb{R}^d$, is sampled. Let $A \in \{0, 1\}$ represent a binary random variable (*r.v.*) indicating whether a participant is assigned to control ($A = 0$) or treatment ($A = 1$). A control or treatment experiment is conducted for $\mathbf{X}$, resulting in the experimental outcomes $Y^A(\mathbf{X})$ as follows,

$$Y^A(\mathbf{X}) = A\Delta(\mathbf{X}) + f(\mathbf{X}) + E, \quad E \sim \mathcal{N}(0, \sigma^2) \tag{1}$$

Here, $E$ represents the noise *r.v.*, and $f(\mathbf{x})$ represents the participants' control outcome without noise, which varies with the covariate $\mathbf{X}$. In contrast, $\Delta(\mathbf{x})$ represents the treatment effect size and contributes to the experimental outcome $Y^A$ only when a participant is assigned to the treatment group, or, $A = 1$. We assume that the outcome-generating model in equation 1 contains *i.i.d* zero-mean Gaussian noise *r.v.* $E$ for participants. However, $E$ is *only required* to follow a zero-mean Gaussian distribution, as in a conventional data model (See Section 13.2 in Wasserman (2013)), and $\sigma^2$ does not need to be identical across participants. This simplification does not affect the validity of our the proposed algorithm in Section 4.2 nor its theoretical analysis in Section 4.3.

### 2.2 A CONVENTIONAL MATCHED-PAIR EXPERIMENTAL DESIGN

One conventional way of forming a matched pair is to *randomly* sample $n$ participant $\{\mathbf{X}_i\}_{i=1}^n$ from a population following the distribution $p_{\mathbf{X}}(\mathbf{x})$. Then, another sequence of participant $\{\mathbf{X}_i'\}_{i=1}^n$ is further identified to pair with $\{\mathbf{X}_i\}_{i=1}^n$, ensuring a sufficiently small distance between $\mathbf{X}_i$ and $\mathbf{X}_i', \forall i \in [1, n]$. This results in the matched pairs $\{(\mathbf{X}, \mathbf{X}')_i\}_{i=1}^n$. An experimenter randomly

assigns the left unit in each pair $(\mathbf{X}, \mathbf{X}')$ to $A$ (treatment or control), and the right unit to the opposite $1 - A$. Herein, we denote the experimental data collected for the $n$ pairs of participants as $\mathcal{F}_n = \{((\mathbf{O}, A), (\mathbf{O}', 1 - A))_i\}_{i=1}^n$ where $\mathbf{O}$ and $\mathbf{O}'$ represent $(\mathbf{X}, Y^A(\mathbf{X}))$ and $(\mathbf{X}', Y^{1-A}(\mathbf{X}'))$, respectively. The experimenter then compares the outcomes $Y^A$ and $Y^{1-A}$ summarized from each matched-pair in $\mathcal{F}_n$ to determine whether the treatment is effective. A two-sample test, such as the $t$-test, is typically performed to make a binary decision about the existence of treatment effect. A key characteristic of this *conventional* design is that the resulting pairs of *r.v.s* $\{(\mathbf{X}, \mathbf{X}')_i\}_{i=1}^n$ are *i.i.d.*. The left covariate unit $\mathbf{X}$ in a pair follows $p_{\mathbf{X}}(\mathbf{x})$, while the right unit $\mathbf{X}'$ approximately follows $p_{\mathbf{X}}(\mathbf{x})$, given that $\mathbf{X}$ and $\mathbf{X}'$ are sufficiently close.

## 2.3 TWO-SAMPLE TESTING

The experimenter conducts a two-sample test on participants' responses gathered from treatment and control groups to determine whether the treatment is effective. Perhaps the most widespread two-sample test is the two-sample $t$-test (Student, 1908). Specially, the two-sample $t$-test evaluates the mean difference: $\frac{1}{n} \sum_{i=1}^n Y_i^1 - \frac{1}{n} \sum_{i=1}^n Y_i^0$, resulting from $\mathcal{F}_n = \{((\mathbf{O}, A), (\mathbf{O}', 1 - A))_i\}_{i=1}^n$, between treatment and control outcomes. The test then determines whether the treatment effect $\mathbb{E}_{\mathbf{X} \sim p_{\mathbf{X}}}[\Delta(\mathbf{X})]$ in equation 1 is larger than 0. In the experimental design considered in this paper, we adopt a more generic two-sample test which determines whether the treatment and control outcome samples are generated from the same distribution. Formally, the experimenter examines the matched pairs in $\mathcal{F}_n$ to test the following null and alternative hypotheses, $H_0$ and $H_1$,

$$H_0 : p_{Y|A}(y \mid 0) = p_{Y|A}(y \mid 1), \quad H_1 : p_{Y|A}(y \mid 0) \neq p_{Y|A}(y \mid 1) \tag{2}$$

where $Y \mid a \equiv Y^a(\mathbf{X})$ and $\mathbf{X} \sim p_{\mathbf{X}}$. Two important metrics for evaluating a two-sample test are:

- Type I error: Indicates the probability of *mistakenly* rejecting $H_0$ when $H_0$ is true.
- Testing power: Indicates the probability of *correctly* rejecting $H_0$ when $H_1$ is true.

The MPED considered in this work involves a *sequential* two-sample testing framework that iteratively processes the experimental data $\mathcal{F}_n$, makes decisions between $H_0$ and $H_1$ and terminates when $H_0$ is rejected. We define $k(\alpha, \mathcal{F}_n)$ as a sequential two-sample testing function that takes a significance level $\alpha \in [0, 1]$ and the data $\mathcal{F}_n$ as input, and outputs a decision variable $v \in \{0, 1\}$, indicating whether to reject $H_0$. A legitimate sequential test is required to satisfy the statistical validity:

**Definition 2.1.** *(Statistical validity for conventional MPED) A sequential test is statistically valid if, under $H_0$, $P(\exists n \geq 1, k_n(\alpha, \mathcal{F}_n) = 1) \leq \alpha, \mathbf{X} \sim p_{\mathbf{X}}$.*

Definition 2.1 states that when participants are randomly enrolled under MPED, a valid sequential test ensures that the Type I error rate is upper-bounded by the significance level $\alpha$. A suite of sequential two-sample tests (Shekhar & Ramdas, 2023; Podkopaev & Ramdas, 2023; Lhéritier & Cazals, 2018) has been developed to preserve such *statistical validity* for *conventional* MPED.

# 3 PROBLEM SETUP

The primary goal of the experimental design considered in this work is to determine between $H_0$ and $H_1$, as defined in 2. This represents a two-sample testing problem aimed at evaluating the distributional equality of participants' responses in the treatment and control groups. Let $\mathcal{X}$ denote the support of $p_{\mathbf{X}}$. We make the following assumption:

**Assumption 3.1.** *(a) Under $H_0$: $\forall \mathbf{x} \in \mathcal{X}, \Delta(\mathbf{x}) = 0$. (b) Under $H_1$: $\forall \mathbf{x} \in \mathcal{X}, \Delta(\mathbf{x}) \geq 0$; moreover, $\exists \gamma > 0$ and $\Omega_\gamma \subset \mathcal{X}$, such that $\forall \mathbf{x} \in \Omega_\gamma, \Delta(\mathbf{x}) \geq \gamma$, and $\mathbb{E}_{\mathbf{X} \sim p_{\mathbf{X} \mid \Omega_\gamma}}[\Delta(\mathbf{X})] > \mathbb{E}_{\mathbf{X} \sim p_{\mathbf{X}}}[\Delta(\mathbf{X})]$[1].*

Under Assumption 3.1, $H_0$ indicates the absence of treatment effect, i.e., $\Delta(\mathbf{x}) = 0, \forall \mathbf{x} \in \mathcal{X}$. $H_1$ states that there exists a region $\Omega_\gamma \subseteq \mathcal{X}$ where the treatment effect exceeds $\gamma$, and that the expected treatment effect over $\Omega_\gamma$ is larger than that over the entire space $\mathcal{X}$. In clinical settings, $\gamma$ *is a user-defined threshold based on prior knowledge, representing the minimum clinically meaningful effect size*. A conventional MPED *randomly* samples participants $\mathbf{X}$ and $\mathbf{X}'$ from a large population to form

---

[1]Clinical trials are conducted in multiple phases. In particular, a treatment that passes Phase I is typically guaranteed not to pose significant harm to patients (Leavitt, 2024), which implies $H_1 : \forall \mathbf{x} \in \mathcal{X}, \Delta(\mathbf{x}) > 0$.

*i.i.d* matched-pairs $\{(\mathbf{X}, \mathbf{X}')_i\}_{i=1}^{n}$, often allocating experimental resources in the unresponsive/low treatment-effect region when $H_1$ is true. Additionally, the region $\Omega_\gamma$ of responders is not known a priori by the experimenter. Therefore, a natural strategy, as suggested in Simon & Simon (2013); Burnett & Jennison (2021); Thall (2021), is to identify the high treatment-effect region $\Omega_\gamma$ through data-driven methods and allocate experimental resources in $\Omega_\gamma$. *We formalize the problem as follows.*

Suppose an experimenter has access to *a large unlabeled population of participants* $\{\mathbf{X}_i\}_{i=1}^{M}$ gathered from $p_{\mathbf{X}}$. Here, "unlabeled" means the experimenter has not conducted any experiments with $\{\mathbf{X}_i\}_{i=1}^{M}$ to acquire experimental outcomes. Additionally, she can sample a participant $\tilde{\mathbf{X}} \in \{\mathbf{X}_i\}_{i=1}^{M}$ and pair it with another $\tilde{\mathbf{X}}' \in \{\mathbf{X}_i\}_{i=1}^{M}$ to form a matched-pair $\left(\tilde{\mathbf{X}}, \tilde{\mathbf{X}}'\right)$ *with negligible cost.* Let $B$ represent the maximum number (or label budget) of the participant pairs $\left(\tilde{\mathbf{X}}, \tilde{\mathbf{X}}'\right)$ that the experimenter can include to perform *expensive* treatment or control experiments to obtain experimental outcomes $\left(Y^A\left(\tilde{\mathbf{X}}\right), Y^{1-A}\left(\tilde{\mathbf{X}}'\right)\right)$. Then, the experimenter pre-selects $\gamma$, which defines the target region $\Omega_\gamma$ (*unknown to the experimenter initially*), and a significance level $\alpha \in [0, 1]$, indicating the Type I error for a two-sample test. She *actively* samples from $\{\mathbf{X}_i\}_{i=1}^{M}$ to form matched-pairs $\left\{\left(\tilde{\mathbf{X}}, \tilde{\mathbf{X}}'\right)\right\}_{i=1}^{n}, n \leq B \ll M$. Meanwhile, the experimenter performs a two-sample test to evaluate the distributional equality of the treatment and control outcomes summarized from $\left\{\left(Y^A\left(\tilde{\mathbf{X}}\right), Y^{1-A}\left(\tilde{\mathbf{X}}'\right)\right)\right\}_{i=1}^{n}$. The experimenter is expected to ensure the following within $B$:

- Under $H_0$, including an active design in MPED still maintains the *validity* of the two-sample test, meaning *Type I error* is less than or equal to $\alpha$.

- Under $H_1$, the active design identifies an enrollment region $\hat{\Omega}_\gamma$ as an approximation of $\Omega_\gamma$, enrolling participants from $\hat{\Omega}_\gamma$ into experiments to increase testing power over conventional MPED.

- Under $H_1$, the enrollment region $\hat{\Omega}_\gamma$ is expected to include sufficient true responders from $\Omega_\gamma$, preventing the false conclusion that the treatment is not broadly applicable.

In addition, we assume that a matching strategy is pre-defined, resulting in *balanced* covariates within each matched pair $\left(\tilde{\mathbf{X}}, \tilde{\mathbf{X}}'\right)$. This is formalized by the following assumption:

**Assumption 3.2.** *For any matched-pair* $\left(\tilde{\mathbf{X}}, \tilde{\mathbf{X}}'\right) \in \mathcal{X} \times \mathcal{X}$, $\left(Y^0, Y^1\right) \perp\!\!\!\perp A \mid \left(\tilde{\mathbf{X}}, \tilde{\mathbf{X}}'\right)$.

Here, $\left(Y^0, Y^1\right)$ represents the corresponding potential treatment and control outcomes, respectively, for $\left(\tilde{\mathbf{X}}, \tilde{\mathbf{X}}'\right) \in \mathcal{X} \times \mathcal{X}$ given a treatment assignment $A$. Assumption 3.2 ensures the unconfoundedness for validating the effectiveness of a treatment in the MPED. A similar assumption is discussed in Section 12.2.2 in Imbens & Rubin (2015). A body of work (Rubin & Thomas, 1996; Heckman et al., 1998; Glazerman et al., 2003; Gelman & Meng, 2004) has focused on ensuring high-quality matching in MPED to support Assumption 3.2. In contrast, our problem setup assumes a pre-defined matching strategy and instead focuses on the sampling strategy for selecting $\tilde{\mathbf{X}}$ from $\{\mathbf{X}_i\}_{i=1}^{M}$. In what follows, we abbreviate $\{a_i\}_{i=1}^{n}$ to $(a)^n$. We write $\left(\tilde{\mathbf{X}}, \tilde{\mathbf{X}}'\right)^n$ to represent a generic sequence of pairs which can be *non-i.i.d., i.i.d. or mixture of both*, while $(\mathbf{X}, \mathbf{X}')^n$ represents only *i.i.d..* pairs.

## 4 MATCHED-PAIR EXPERIMENTAL DESIGN WITH ACTIVE LEARNING

This section formalizes the identification of $\Omega_\gamma$ as an active learning problem and provides a theoretical analysis. *Practitioners may safely proceed to Section 5, which presents a practical instantiation.*

### 4.1 FINDING $\Omega_\gamma$ WITH ACTIVE LEARNING

In Figure 2, we formalize the identification of $\Omega_\gamma$ as an *active learning* problem. *Active learning framed within MPED aims to acquire a classifier to identify $\Omega_\gamma$ with a limited label budget.* Figure 2 presents a problem for constructing a classifier by actively labeling $\tilde{\mathbf{X}}$ under a label budget $B$. In this

Suppose $(\mathbf{X})^M$ is *i.i.d.* sampled from $p_{\mathbf{X}}$, and an experimenter conducts treatment and control experiment using a matched-pair $\left(\tilde{\mathbf{X}}, \tilde{\mathbf{X}}'\right)$, resulting in outcomes $\left(Y^A\left(\tilde{\mathbf{X}}\right), Y^{1-A}\left(\tilde{\mathbf{X}}'\right)\right)$. She then labels $\tilde{Z}$ of $\tilde{\mathbf{X}}$ as 1 if $Y^1\left(\tilde{\mathbf{X}}\right) - Y^0\left(\tilde{\mathbf{X}}'\right) \geq \gamma$ $\left(\text{resp. } Y^1\left(\tilde{\mathbf{X}}'\right) - Y^0\left(\tilde{\mathbf{X}}\right) \geq \gamma\right)$, or as 0 otherwise. The active learning for MPED involves, given a label budget $B$, constructing $\left(\tilde{\mathbf{X}}, \tilde{\mathbf{X}}'\right)^n$ from $(\mathbf{X})^M$, and, experimenting on the matched-pairs to obtain labeled data $\left(\tilde{\mathbf{X}}, \tilde{Z}\right)^n$. The goal is to construct a classifier function $q: \mathbb{R}^d \to \{0, 1\}$ with respect to $p_{\mathbf{X}\tilde{Z}}$, using $\left(\tilde{\mathbf{X}}, \tilde{Z}\right)^n$ subject to $n \leq B \ll M$.

Figure 2: Active learning framed under MPED. "Resp." is an abbreviation for "respectively".

setup, the feature and label variables represent the the participant covariate, and, an binary indicator which denotes whether the treatment effect exceeds $\gamma$ within a pair of experimental outcomes $\left(Y^A\left(\tilde{X}\right), Y^{1-A}\left(\tilde{X}\right)\right)$. Consequently, the following proposition holds:

**Proposition 4.1.** *Under $H_1$, Assumption 3.2 and given $p_{\mathbf{X}\tilde{Z}}$, consider the Bayes optimal classifier defined as* $q^*(\mathbf{x}) = \begin{cases} 1 \text{ if } P_{\tilde{Z}|\mathbf{X}}(1 \mid \mathbf{x}) \geq 0.5 \\ 0 \text{ otherwise} \end{cases}$ *. Then we have* $\Omega_\gamma = \{\mathbf{x} \in \mathcal{X} \mid q^*(\mathbf{x}) = 1\}$.

Assumption 3.2 ensures that for any $(\tilde{\mathbf{x}}, \tilde{\mathbf{x}}') \in \mathcal{X}^2$, we have $p(y^a \mid \tilde{\mathbf{x}}) = p(y^a \mid \tilde{\mathbf{x}}')$, yielding perfect identification of the label $\tilde{z}$ in MPED. Thus, $P_{\tilde{Z}|\mathbf{X}}$ exactly captures the probability that the treatment outcome exceeds the control outcome by at least $\gamma$ for a given $\mathbf{x}$. Hence, finding the Bayes classifier with respect to $p_{\mathbf{X}\tilde{Z}}$ is *sufficient* for identifying the target $\Omega_\gamma$. Details are provided in Appendix C. **Practicality of perfect label identification**: In practice, experimenters define baseline covariates such as gender, race, or other variables and generate matched pairs conditional on these covariates ((Bai, 2022), p. 3912). In this setting, the label identification process in Figure 2 is effectively perfect for the baseline covariates, yielding a classification problem in their space. Thus, assuming perfect label identification for $\tilde{\mathbf{x}}$ does not reduce the practical relevance of our approach.

### 4.2 THE MATCHED-PAIR EXPERIMENTAL DESIGN WITH ACTIVE LEARNING

Our experimental design relies on the *RobustCAL* algorithm detailed in Section 5.2 of Hanneke et al. (2014). *RobustCAL* is a variant of an agnostic active learning algorithm proposed in Balcan et al. (2006), where the term "agnostic" indicates that *RobustCAL* is robust to classification noise. Specifically, the typical data model described in equation 1 assumes that the experimental outcomes contain noise, which leads to an agnostic active learning problem in Figure 2. Accordingly, we propose the *matched-pair experimental design with RobustCAL (MPED-RobustCAL)* in Algorithm 1. The core of *MPED-RobustCAL* lies in Line 4, which actively labels points from DIS($\mathcal{C}$)—the region where the classifier set $\mathcal{C}$ disagrees—to efficiently reduce classification error when predicting $\Omega_\gamma$. Additionally, it labels points from POS($\mathcal{C}$), which consists of points classified by $\mathcal{C}$ as belonging to $\Omega_\gamma$, to enhance the testing power for MPED. The function $k$ denotes a *sequential* two-sample testing procedure that evaluates the collected experimental data $\tilde{\mathcal{F}}$ using the significance level $\alpha$. *MPED-RobustCAL* terminates when the test $k$ returns $v = 1$ indicating that $H_0$ is rejected (i.e., the treatment is deemed effective), or when the label budget $B$ is exhausted.

**Notations in *MPED-RobustCAL*** $B$ represents the label budget, $\delta$ denotes the failure probability of the algorithm, $\alpha$ represents the significance level for a two-sample test $k$ and $\gamma$ indicates the treatment effect threshold for defining the target region $\Omega_\gamma$. $\mathbb{C} = \left\{q \mid q: \mathbb{R}^d \to \{0, 1\}\right\}$ denotes the original class of classification functions, from which an analyst searches for a classifier to identify $\Omega_\gamma$. The total number of currently generated $\tilde{\mathbf{X}}$ is denoted by $m$, and $Q$ represents the set of features and queried label pairs, while $\tilde{\mathcal{F}}$ indicates a set of available experimental data, including elements $\left(\left(\tilde{\mathbf{O}}, A\right), \left(\tilde{\mathbf{O}}', 1 - A\right)\right)$, where $\tilde{\mathbf{O}} = \left(\tilde{\mathbf{X}}, Y^A\left(\tilde{\mathbf{X}}\right)\right)$ and $\tilde{\mathbf{O}}' = \left(\tilde{\mathbf{X}}', Y^{1-A}\left(\tilde{\mathbf{X}}'\right)\right)$. DIS$(\mathcal{C}) = \{\mathbf{x} \in \mathcal{X} \mid \exists h, q \in \mathcal{C}, \text{ s.t. } h(\mathbf{x}) \neq q(\mathbf{x})\}$ includes points where classification functions in

---

**Algorithm 1** MPED-RobustCAL$_\delta$ $(B, \alpha, \gamma)$

---

1: $m \leftarrow 0, Q \leftarrow \{\}, \tilde{\mathcal{F}} \leftarrow \{\}, \mathcal{C} \leftarrow \mathbb{C}$
2: **while** $|Q| < B$ and $m < 2^B$ **do**
3:      $m \leftarrow m + 1$
4:      **if** $\tilde{\mathbf{X}} \in \hat{\Omega}_\gamma = \text{DIS}(\mathcal{C}) \bigcup \text{POS}(\mathcal{C})$ **then**
5:          Form a matched-pair $\left(\tilde{\mathbf{X}}, \tilde{\mathbf{X}}'\right)$ and randomly assign them to treatment/control experi-
         ments leading to $\left(Y^A\left(\tilde{\mathbf{X}}\right), Y^{1-A}\left(\tilde{\mathbf{X}}'\right)\right)$
6:          Request label $\tilde{Z}$ of $\tilde{\mathbf{X}}$ using $\gamma$ as described in Figure 2
7:          $Q \leftarrow Q \bigcup \left\{\left(\tilde{\mathbf{X}}, \tilde{Z}\right)\right\}; \tilde{\mathcal{F}} \leftarrow \tilde{\mathcal{F}} \bigcup \left\{\left(\left(\tilde{\mathbf{O}}, A\right), \left(\tilde{\mathbf{O}}', 1 - A\right)\right)\right\}$
8:          $v = k\left(\alpha, \tilde{\mathcal{F}}\right)$; **if** $v == 1$ **then break**
9:      **if** $\log_2(m) \in \mathbb{N}$ **then** $\mathcal{C} \leftarrow \{q \in \mathcal{C} \mid (\text{er}_Q(q) - \min_{h \in \mathcal{C}} \text{er}_Q(h)) |Q| \leq \bar{U}(m, \delta) m\}$
10: **return** $\mathcal{C}$ and $v \in \{0, 1\}$

---

$\mathcal{C} \subseteq \mathbb{C}$ disagree with, while $\text{POS}(\mathcal{C}) = \{\mathbf{x} \in \mathcal{X} \mid \forall q \in \mathcal{C}, q(\mathbf{x}) = 1\}$ represents points predicted as 1 by all classifiers in $\mathcal{C}$. The empirical risk of a classifier $q$ over the labeled set $Q$ is denoted by $\text{er}_Q(q)$, and $\bar{U}$ is a predefined function used to eliminate poorly performing classifiers from $\mathbb{C}$. Additionally, *MPED-RobustCAL* incorporates the sequential two-sample testing function $k$ to decide whether to reject $H_0$, resulting in a *decision variable* $v \in \{0, 1\}$, where $v = 1$ indicates that $H_0$ is rejected.

**How does *MPED-RobustCAL* work?** Central to *MPED-RobustCAL* are the steps highlighted in blue in Algorithm 1. Unlike RobustCAL (Hanneke et al., 2014), *MPED-RobustCAL* queries labels for features beyond DIS $(\mathcal{C})$ and incorporates seqential two-sample testing $k$. Compared to *passive learning*[2], *MPED-RobustCAL* selectively queries the label of $\tilde{\mathbf{X}}$ sampled from $p_{\mathbf{X}}$ *only if* $\tilde{\mathbf{X}}$ belongs to the union of the disagreement region DIS $(\mathcal{C})$ and the positive region POS $(\mathcal{C})$. This approach results in a classifier with the same classification error but requires fewer label queries than passive learning. The efficiency arises because *MPED-RobustCAL* prioritizes querying labels for $\tilde{\mathbf{X}} \in \text{DIS}(\mathcal{C})$, where classifiers in $\mathcal{C}$ disagree, leading to the elimination of a similar number of classifiers in $\mathcal{C}$ with fewer labels compared to passive learning. This classifier elimination is detailed in Line 9, where classifiers with empirical risks larger than the smallest empirical risk by a margin determined by the pre-defined function $\bar{U}$ are eliminated. Additionally, *MPED-RobustCAL* prioritizes label querying in the positive region POS $(\mathcal{C})$, as it aims to enroll participants in $\Omega_\gamma$ to enhance the label efficiency of the sequential test $k$ in rejecting $H_0$ under $H_1$. Finally, the algorithm returns $v$, indicating whether to reject $H_0$, and $\mathcal{C}$, which is used to acquire the enrollment region $\hat{\Omega}_\gamma$.

**Remark 4.2.** *The choice of $\bar{U}(m, \delta)$ for MPED-RobustCAL is identical to that in RobustCAL (Hanneke et al., 2014). we refer readers to equation 20 in Appendix for its expression.*

**Clinical Implications of *MPED-RobustCAL*** As illustrated in Figure 1 and also guaranteed by Theorem 4.5 in Section 4.3, *MPED-RobustCAL* consistently enrolls participants from the region $\hat{\Omega}_\gamma = \text{DIS}(\mathcal{C}) \cup \text{POS}(\mathcal{C})$ into experiments, ensuring that the enrollment region *encloses* the target region $\Omega_\gamma$. This enclosing property of *MPED-RobustCAL* provides the unique benefit of identifying all responders in $\Omega_\gamma$. This stands in stark contrast with existing active designs, which may miss many responders in the target region $\Omega_\gamma$, leading to the false conclusion that the treatment is not broadly applicable and thereby to the premature termination of follow-up studies.

## 4.3 LABEL COMPLEXITY OF *MPED-RobustCAL*

Let $d_{\text{vc}}$ denote the *Vapnik-Chervonenkis* (VC) dimension of the classifier class $\mathbb{C}$. The VC dimension, $d_{\text{vc}}$, quantifies the complexity of the classifier class $\mathbb{C}$, effectively reflecting the "size" of $\mathbb{C}$ from which an optimal classifier can be selected. We refer readers to Vapnik & Chervonenkis (2015) or D.4 in Appendix for details. Additionally, we introduce a definition concerning the structure of $p_{\mathbf{X}}$.

---

[2]Here, passive learning refers to querying the label of every $\mathbf{X}$ generated from $p_{\mathbf{X}}$ to update $\mathcal{C}$.

**Definition 4.3.** *(Disagreement Coefficient $\theta_q(r_0)$ (Hanneke et al., 2014)) Given a classifier $q \in \mathbb{C}$ and a probability constant $r \in [0, 1]$, we write $B(q, r) = \{h \in \mathbb{C} \mid P(h(\mathbf{X}) \neq q(\mathbf{X})) \leq r, \mathbf{X} \sim p_{\mathbf{X}}\}$ to represent a class of classifiers whose label predictions disagree with $q$ with the probability $r$ at most. Then, $\forall r_0 \geq 0$, define the disagreement coefficient of $q$ with respect to $\mathbb{C}$ under $p_{\mathbf{x}}$ as $\theta_q(r_0) = \sup_{r > r_0} \frac{p_{\mathbf{x}}(DIS(B(q,r)))}{r} \bigvee 1$.*

$\theta_q(r_0)$ characterizes the probability that a point $\mathbf{X} \sim p_{\mathbf{X}}$ resides in the disagreement region DIS $(\mathcal{C})$. As stated in Section 4.2, only labeling points in DIS $(\mathcal{C})$ contributes to eliminating poorly performing classifiers from $\mathcal{C}$. Therefore, a smaller $\theta_q(r_0)$ indicates a more significant improvement of *MPED-RobustCAL* over passive learning, as the latter wastes many labels on points outside DIS $(\mathcal{C})$.

Recall that $P_{\tilde{Z}|\mathbf{X}}$ results from the labeling process illustrated in Figure 2. We define $\tilde{\eta}(\mathbf{x}) = P_{\tilde{Z}|\mathbf{X}}\left(\tilde{Z} = 1 \mid \mathbf{x}\right)$ relative to $p_{\mathbf{X}\tilde{Z}}$ and use $q^*$ to denote the Bayes optimal classifier with respect to $p_{\mathbf{X}\tilde{Z}}$. Lastly, we introduce an assumption regarding the noise of $\tilde{Z}$ with respect to $p_{\mathbf{X}\tilde{Z}}$.

**Assumption 4.4.** *(Bounded noise (Massart & Nedelec, 2006)) Under $H_1$, $\exists a \in [1, \infty)$ such that $P(\mathbf{X} : |\tilde{\eta}(\mathbf{X}) - 1/2| < 1/(2a)) = 0$ where $\mathbf{X} \sim p_{\mathbf{X}}$, and the Bayes optimal classifier $q^* \in \mathbb{C}$.*

$a$ in Assumption 4.4 indicates how noisy $\tilde{Z}$ is, implicitly characterizing the lowest error rate achievable by the Bayes optimal classifier. *MPED-RobustCAL* enrolls participants from DIS $(\mathcal{C}) \bigcup$ POS $(\mathcal{C})$, which fully covers the target $\Omega_\gamma$. The following theorem establishes that $\Omega_\gamma \subseteq \hat{\Omega}_\gamma =$ DIS $(\mathcal{C}) \bigcup$ POS $(\mathcal{C})$ with high probability. Furthermore, the ***ratio of the enrollment region over target region*** $,\mathcal{R} = \frac{|\hat{\Omega}_\gamma|}{|\Omega_\gamma|}$, converges to 1 faster than passive learning, i.e., $\Omega_\gamma$ is efficiently identified.

**Theorem 4.5.** *Under $H_1$ and $p_{\mathbf{X}\tilde{Z}}$ along with Assumption 4.4 and 3.2, let $P(\Omega_\gamma) = P(\mathbf{X} \in \Omega_\gamma), \mathbf{X} \sim p_{\mathbf{X}}$. Passive learning attains a classifier set $\mathcal{C}$ such that $\epsilon = \max_{q \in \mathcal{C}} P(q(\mathbf{X}) \neq q^*(\mathbf{X}))$, and, $\Omega_\gamma \subseteq \hat{\Omega}_\gamma = DIS(\mathcal{C}) \bigcup POS(\mathcal{C})$ with $\mathcal{R} = 1 + \frac{\theta_{q^*}(0)\epsilon}{P(\Omega_\gamma)}$, with probability at least 1 - $\delta$ using a label complexity of*

$$\Lambda'(\epsilon, \delta) = \mathcal{O}\left(\frac{1}{\epsilon}\left(d_{vc} \log(\theta_{q^*}(0)) + \log(1/\delta)\right)\right). \tag{3}$$

*In contrast, to attain the same result with probability at least $1 - \delta$, the MPED-RobustCAL requires a label complexity of*

$$\Lambda'(\epsilon, \delta) P(\Omega_\gamma) + \Lambda(\epsilon, \delta), \tag{4}$$

*in which $\Lambda(\epsilon, \delta) = \mathcal{O}\left(\log\left(\frac{1}{\epsilon}\right)\theta_{q^*}(0) \times \left(d_{vc} \log(\theta_{q^*}(0)) + \log\left(\frac{\log(1/\epsilon)}{\delta}\right)\right)\right)$.*

**Remark 4.6.** *equation 4 indicates a fractional decrease in label complexity compared to equation 3, suggesting that the ratio $\mathcal{R}$ for MPED-RobustCAL converges to 1 faster than that for passive learning. However, this convergence rate is slower than that of the original RobustCAL. This slowdown arises because MPED-RobustCAL queries additional labels from POS($\mathcal{C}$), aiming to efficiently detect the existence of a treatment effect. Yet, in scenarios where $P(\Omega_\gamma)$ is sufficiently small, $\Lambda$ dominates the label complexity in equation 4, and MPED-RobustCAL recovers the convergence rate of RobustCAL.*

## 5 INSTANTIATION OF *MPED-RobustCAL*

Algorithm 1 facilitates a rigorous theoretical analysis, but it may not be directly applicable for algorithmic implementation. In this section, we provide a practical instantiation of *MPED-RobustCAL*. One of the most conventional implementations of active learning is the query-by-committee (Seung et al., 1992): Given a set of classifiers trained on the current labeled dataset, an active learner selects a point on which the classifiers disagree to query its label. The final prediction is then made by averaging class prediction probabilities of all classifiers. Consequently, DIS $(\mathcal{C})$ in Algorithm 1 is realized by the region where the classifier committee disagrees. However, beyond querying labels in DIS $(\mathcal{C})$ to efficiently train a classifier, *MPED-RobustCAL* also queries labels in POS $(\mathcal{C})$ to facilitate the two-sample testing. To this end, we propose practical *MPED-RobustCAL* in Algorithm 2.

Algorithm 2 takes inputs a label budget $B$, a significance level $\alpha$, and the treatment effect threshold $\gamma$. The classifier set $\mathcal{C}$ is initialized using a small training set $Q_0$, which is obtained through random

label querying from $\mathcal{S}$. *These labeled points are excluded from $\mathcal{S}$ before proceeding to the "active query" starting from Line 4.* The "active query" set $\mathcal{E}$ is defined as a set of unlabeled points for which at least one classifier in $\mathcal{C}$ predicts class one. This indicates that the practical *MPED-RobustCAL* queries labels from the positive region, i.e., unlabeled points predicted as one by all classifiers, and from disagreement regions, i.e., points where at least one classifier predicts one. If $\mathcal{E}$ is empty, the algorithm switches to random label querying. $\mathcal{C}$ is updated whenever new $\left(\tilde{\mathbf{X}}, \tilde{Z}\right)$ and $\left(\tilde{\mathbf{X}}', \tilde{Z}\right)$ are added to the queried set $Q$. Additionally, a sequential test $k$ uses the experimental data $\tilde{\mathcal{F}}$ to decide whether to reject $H_0$. The algorithm terminates when $k$ outputs $v = 1$ or the label budget is exhausted. Otherwise, the classifiers in $\mathcal{C}$ are updated with $Q$ for the next round of experimentation. The outputs of the algorithm are a decision variable $v$ and a classifier set $\mathcal{C}$ used to define the enrollment set $\mathcal{E}$.

---

**Algorithm 2** Practical MPED-RobustCAL $(B, \alpha, \gamma)$

---

1: $\tilde{\mathcal{F}} \leftarrow \{\}, \mathcal{S} \leftarrow (\mathbf{X})^M, Q \leftarrow Q_0$
2: Initialize a set of classifier $\mathcal{C} = \{q(\mathbf{x})\}$ with $Q$; $\mathcal{E} \leftarrow \{\mathbf{X} \in \mathcal{S} \mid q(\mathbf{X}) = 1, \exists q \in \mathcal{C}\}$
3: **while** $|Q| < B$ **do**
4:     **if** $\mathcal{E} \neq \emptyset$ **then** Randomly acquire an $\tilde{\mathbf{X}} \in \mathcal{E}$ **else** Randomly acquire an $\tilde{\mathbf{X}} \in \mathcal{S}$
5:     Form a matched pair $\left(\tilde{\mathbf{X}}, \tilde{\mathbf{X}}'\right)$ and randomly assign them to treatment/control experiments
    leading to $\left(Y^A\left(\tilde{\mathbf{X}}\right), Y^{1-A}\left(\tilde{\mathbf{X}}'\right)\right)$
6:     Request label $\tilde{Z}$ of $\tilde{\mathbf{X}}$ and $\tilde{\mathbf{X}}'$ using $\gamma$ as described in Figure 2
7:     $Q \leftarrow Q \cup \left\{\left(\tilde{\mathbf{X}}, \tilde{Z}\right), \left(\tilde{\mathbf{X}}', \tilde{Z}\right)\right\}$; $\tilde{\mathcal{F}} \leftarrow \tilde{\mathcal{F}} \cup \left\{\left(\left(\tilde{\mathbf{O}}, A\right), \left(\tilde{\mathbf{O}}', 1 - A\right)\right)\right\}$
8:     $v = k\left(\alpha, \tilde{\mathcal{F}}\right)$; **if** $v == 1$ **then break**
9:     Update $\mathcal{C}$ with $Q$; $\mathcal{S} \leftarrow \mathcal{S} \backslash \{\tilde{\mathbf{X}}, \tilde{\mathbf{X}}'\}$; $\mathcal{E} \leftarrow \{\mathbf{X} \in \mathcal{S} \mid q(\mathbf{X}) = 1, \exists q \in \mathcal{C}\}$
10: **return** $\mathcal{C}$ and the decision $v \in \{0, 1\}$.

---

**Statistical Validity of the Two-Sample Test Under Algorithm 2**    $k$ in Algorithm 2 represents a sequential two-sample test that repeatedly examines $\tilde{\mathcal{F}}$, which consists of data generated through *active enrollment*. Definition 2.1 specifies the statistical validity of a sequential test under the *conventional* MPED, where each participant $\mathbf{X}$ is enrolled *randomly*. In this context, we index $\tilde{\mathcal{F}}$ as $\tilde{\mathcal{F}}_n = \left\{\left(\left(\tilde{\mathbf{O}}, A\right), \left(\tilde{\mathbf{O}}', 1 - A\right)\right)_i\right\}_{i=1}^n$, where $n \in [1, B]$ indicates that the $n$-th matched-pair $\left(\tilde{\mathbf{X}}, \tilde{\mathbf{X}}'\right)_n$ is formed, and their corresponding experimental outcomes are included in $\tilde{\mathcal{F}}$. At each $n$, the testing function $k$ utilizes the significance level $\alpha$ and the data $\tilde{\mathcal{F}}_n$ to test between $H_0$ and $H_1$.

**Theorem 5.1.** *(Statistical validity) Suppose an experimenter instantiates $k$ using a statistically valid test as defined in 2.1. Then, under $H_0$, $P\left(\exists n \geq 1, k\left(\alpha, \tilde{\mathcal{F}}_n\right) = 1\right) \leq \alpha$ for MPED-RobustCAL.*

## 6   SIMULATION RESULTS

**Data Description**    We simulate a **synthetic dataset** of 1000 matched pairs, where each participant has covariates $\mathbf{X} = (X_1, X_2)$ drawn independently from Uniform $[0, 1]$. Under $H_1$, a treatment effect $\Delta(\mathbf{X}) = 1$ is applied when $X_1 + s < X_2$ with $s = 0.5$, and zero otherwise; under $H_0$, $\Delta(\mathbf{X}) = 0$ everywhere. Gaussian noise with variance $\sigma^2 = 0.1$ is added to the responses. We also evaluate on two real-world datasets under $H_1$: PRO-ACT (Atassi et al., 2014) ($\sim$770 matched pairs, 9 covariates from ALS clinical trials on Riluzole) and IHDP (Shalit et al., 2017) ($\sim$750 matched pairs, 25 covariates for assessing home-visit effects on cognitive outcomes).

**Implementation Details**    We implement Algorithm 2 to actively enroll participants from $\mathcal{S} = (\mathbf{X})^M$. The testing function $k$ follows the sequential predictive test of Podkopaev & Ramdas (2023). The labeled set $Q$ is bootstrapped into 10 subsets to initialize or update an ensemble $\mathcal{C}$ of 10 classifiers for the synthetic, PRO-ACT, and IHDP datasets. For the simulation results reported in the main paper, we use logistic-regression ensembles for the synthetic data and decision-tree ensembles for the two real-world datasets. The initial labeled set $Q$ contains 50 random samples for the synthetic data and 10 for PRO-ACT and IHDP. We set $\alpha = 0.05$ and use treatment-effect thresholds $\gamma = 0.2, 0.1$, and

4.5 for the synthetic, PRO-ACT, and IHDP datasets, respectively. Additional details on $k$ and full experimental and sensitivity analyses appear in Appendix B.

Table 1: A comparison of the *testing power* between the conventional MPED and *MPED-RobustCAL*.

(a) Synthetic

| Label budget | 200 | 300 | 400 | 500 | 600 | 700 |
|---|---|---|---|---|---|---|
| Conventional | 0.07 | 0.11 | 0.15 | 0.18 | 0.19 | 0.22 |
| *MPED-RobustCAL* | **0.16** | **0.34** | **0.61** | **0.76** | **0.85** | **0.85** |

(b) PRO-ACT

| Label budget | 250 | 300 | 350 | 400 |
|---|---|---|---|---|
| Conventional | 0.10 | 0.29 | 0.40 | 0.67 |
| *MPED-RobustCAL* | **0.19** | **0.39** | **0.59** | **0.79** |

**Evaluations of Testing Power and Type I error** Table 1 presents the testing power of the conventional MPED and *MPED-RobustCAL* resulting from 100 runs. As shown, *MPED-RobustCAL* achieves a higher testing power of rejecting $H_0$ under $H_1$. This

Table 2: Type I error by *MPED-RobustCAL* for synthetic data; $\alpha = 0.05$.

| Label budget | 200 | 300 | 400 | 500 | 600 | 700 |
|---|---|---|---|---|---|---|
| Type I | 0.038 | 0.044 | 0.046 | 0.046 | 0.046 | 0.046 |

improvement results from *MPED-RobustCAL* actively enrolling participants from high treatment-effect regions. Theorem 5.1 implies that the Type I error of practical *MPED-RobustCAL* is still upper-bounded by $\alpha$, even the participants are actively enrolled. Table 2 demonstrate this, showing that the Type I errors are all smaller than $\alpha = 0.05$ on various label budget.

**Evaluations of the True Positve Rate (TPR)** Theorem 4.5 implies that *MPED-RobustCAL* yields an enrollment region that *encloses* the target region $\Omega_\gamma$ of high treatment-effect points. To evaluate this, we present the results for TPR, defined as the ratio of the number of points $\mathbf{x}$ with labels 1 (i.e., points with high treatment effects) included in the enrollment region to the total number of points with labels 1. We also compare *MPED-RobustCAL* with two active designs from Simon & Simon

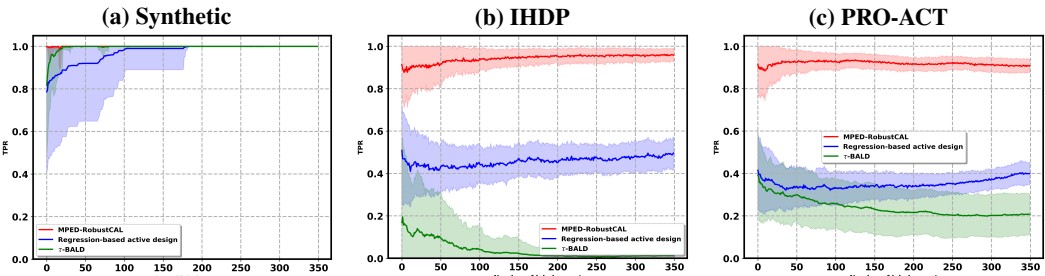

Figure 3: A comparison of the TPR among *MPED-RobustCAL*, the regression-based active design (Simon & Simon, 2013), and $\tau$-BALD (Jesson et al., 2021).

(2013) and Jesson et al. (2021). The design in Simon & Simon (2013) constructs separate regressors $f_t(\mathbf{x})$ and $f_c(\mathbf{x})$ and enrolls points satisfying $f_t(\mathbf{x}) - f_c(\mathbf{x}) \geq \gamma$; we refer to this method as the *regression-based active design*. The work in Jesson et al. (2021) describes an approach based on Bayesian active learning by disagreement (BALD), termed $\tau$-*BALD*, which actively labels samples to approximate the effect size $\Delta(\mathbf{x})$ in equation 1 with a single regressor $g(\mathbf{x})$ and enrolls points with $g(\mathbf{x}) \geq \gamma$. For the regression-based active design, we use a Gaussian process for the synthetic data and decision trees for the two real-world datasets; for $\tau$-BALD, we use a Gaussian process to perform Bayesian active learning. We obtain the TPR for the enrollment regions identified by *MPED-RobustCAL*, the regression-based active design, and $\tau$-BALD by evaluating 100 validation sets. Figure 3 shows that *MPED-RobustCAL* consistently achieves a higher TPR, indicating that it includes more participants from the target region than the two existing active designs.

## 7 CONCLUSION

We propose an innovative MPED framework that actively enrolls participants from regions with high treatment effects. Our approach formulates the identification of responsive regions as a classification task, leading to the algorithm *MPED-RobustCAL*. Theoretical analysis shows that the resulting enrollment region not only encloses but also converges to the true responsive region, achieving a fractional improvement in label complexity compared to passive learning. Experimental results on both synthetic and real-world datasets validate the advantages of our proposed design over both the conventional MPED and the existing active designs.

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

# A  THE USE OF LARGE LANGUAGE MODELS (LLMS)

The writing of this submission was polished with the assistance of a large language model (LLM) tool.

# B  FULL EXPERIMENTAL RESULTS

This section presents the complete simulation results of experiments conducted on synthetic data, PRO-ACT (Atassi et al., 2014), and IHDP (Shalit et al., 2017), along with their implementation details.

## B.1  INSTANTIATION OF THE SEQUENTIAL TWO-SAMPLE TEST $k$

*MPED-RobustCAL* detects treatment effectiveness using two-sample testing. This section introduces a specific two-sample test—a sequential predictive test based on betting—to instantiate the sequential testing function $k$ in *MPED-RobustCAL* used in both Algorithm 1 and 2.

**Sequential two-sample testing**  Recalling the formulation of two-sample testing in Section 2.3, we denote $(\mathbf{S}, A)$ as the feature and label random variables, where $(\mathbf{S}, A) \sim p_{\mathbf{S}A}(\mathbf{s}, a)$. For example, $p_{\mathbf{S}A}(\mathbf{s}, a)$ can represent the joint distribution of participants' responses and their treatment/control assignments in a *conventional* MPED that randomly enrolls participants. A sequential test receives observations of $(\mathbf{S}, A)$ one at a time and determines, upon each arrival, whether to accept or reject the null $H_0 : p_{\mathbf{S}|A}(\mathbf{s} \mid 0) = p_{\mathbf{S}|A}(\mathbf{s} \mid 1)$ against the alternative $H_1 : p_{\mathbf{S}|A}(\mathbf{s} \mid 0) \neq p_{\mathbf{S}|A}(\mathbf{s} \mid 1)$.

**Sequential predictive two-sample testing Podkopaev & Ramdas (2023)**  *Testing by betting* has been extensively discussed in Shekhar & Ramdas (2023); Shafer (2021), capturing the following idea: Under $H_0$, a bettor will neither gain or lose wealth regardless of the betting strategy. In contrast, under $H_1$ and with an appropriate betting strategy, the bettor's wealth will grow rapidly, indicating the bet is profitable (i.e., $H_1$ is true). Podkopaev & Ramdas (2023) introduces a sequential predictive two-sample test based on the betting. We present this test as follows.

---

**Sequential predictive test based on betting**: Given an initial statistic (or wealth) $W_0 = 1$ and a significance level $\alpha \in [0, 1]$, an experimenter begins at $n = 1$ and sequentially receives $(\mathbf{S}, A)_n, n \geq 1$, where $(\mathbf{S}_n, A_n) \sim p_{\mathbf{S}A}$. The experimenter updates the statistic (or wealth) sequentially whenever a new $(\mathbf{S}, A)$ arrives by

$$W_n = W_{n-1}\left(1 + \lambda_n L_n\left(\mathbf{S}_n, A_n\right)\right)$$

$$= \prod_{i=1}^{n}\left(1 + \lambda_i L_i\left(\mathbf{S}_i, A_i\right)\right) \tag{5}$$

in which, $L_n(\mathbf{S}, A) = (2A_n - 1)(2\bar{q}_n(\mathbf{S}_n) - 1)$ represents the payoff function, and $\lambda_n \in [-1, 1], \forall n > 0$ denotes betting fraction, both updated sequentially. $\bar{q}_n$ is a classifier developed with respect to $p_{\mathbf{S}A}$ to predict $A$ from $\mathbf{S}$. The experimenter stops the test if $W_n \geq \frac{1}{\alpha}$ to reject $H_0$.

---

The payoff function $L_n(\mathbf{S}_n, A_n)$ in equation 5 returns a value in $\{-1, 1\}$ based on $(\mathbf{S}_n, A_n)$. It is updated sequentially through *online learning* of the classifier $\bar{q}_n$. Assuming $\lambda_n, \forall n > 0$ are positive, if $\bar{q}_n$ correctly predicts the true label $A_n$, the experimenter wins the bet, and the statistic (or wealth) $W_{n-1}$ increases by $\lambda_n W_{n-1}$. Conversely, if the prediction is incorrect, the experimenter loses the bet, and $W_{n-1}$ decreased by $\lambda_n W_{n-1}$. Under $H_0$, the experimenter is playing a fair game and $W_n$ remains unchanged in expectation. However, under $H_1$, as the classifier $\bar{q}_n$ improves over time and and with an appropriate *betting strategy* for selecting the betting fraction $\lambda$, $W_n$ grows exponentially, leading to the rejection of $H_0$. Shekhar & Ramdas (2023) recommends using the *online Newton step (ONS)* proposed in Cutkosky & Orabona (2018) to sequentially identify $\lambda_i, \forall i > 0$, that maximize $\mathbb{E}_{(\mathbf{S}_i, A_i) \sim p_{\mathbf{S}A}}\left[\log\left(1 + \lambda_i L_i\left(\mathbf{S}_i, A_i\right)\right)\right]$ under $H_1$. We refer readers to Definition 5 in Shekhar & Ramdas (2023) for details of the ONS.

**Applying the sequential predictive test to *MPED-RobustCAL*** Algorithm 3 elaborates on instantiating the sequential test $k$ in (practical) *MPED-RobustCAL* with the predictive test (Podkopaev & Ramdas, 2023). For clarity in the subsequent presentation, we index $k$ by $n \in [1, B]$, where $n$ denotes the $n$-th matched-pair $\left(\tilde{\mathbf{X}}, \tilde{\mathbf{X}}'\right)_n$ formed and included in the experiment. Accordingly, we also index the $n$-th experimental data as $\left(\left(\tilde{\mathbf{O}}, A\right), \left(\tilde{\mathbf{O}}', 1-A\right)\right)_n$. Consequently, the sequential testing function $k_n$ utilizes the past experimental data $\mathcal{F}_{n-1}$ and the latest experimental data to decide between $H_0$ and $H_1$. Specifically, *only one unit of the latest matched-pair, e.g.,* $\left(\tilde{\mathbf{O}}, A\right)_n$, is used here. $\mathcal{F}_{n-1}$ consists of $\left(\left(\tilde{\mathbf{O}}, A\right), \left(\tilde{\mathbf{O}}', 1-A\right)\right)^{n-1}$ ($\mathcal{F}_0 = \emptyset$). The experimenter constructs

---

**Algorithm 3** Predictive test $k_n\left(\alpha, \mathcal{F}_{n-1}, \left(\tilde{\mathbf{O}}, A\right)_n\right)$

---

1: Update/Initialize a classifier $\bar{q}_n$ using $\mathcal{F}_{n-1}$
2: $\lambda_{n+1} \leftarrow \text{ONS}\left(\lambda_n, \left(\tilde{\mathbf{O}}, A\right)_n\right)$
3: $W_n \leftarrow W_{n-1}\left(1 + \lambda_n L_n\left(\tilde{\mathbf{O}}_n, A_n\right)\right)$ to $W_n$
4: **if** $W_t \geq \frac{1}{\alpha}$, **then return** $v \leftarrow 1$ **else return** $v \leftarrow 0$

---

a classifier $\bar{q}_t$ using both $\left(\tilde{\mathbf{O}}\right)^{n-1}$ and $\left(\tilde{\mathbf{O}}'\right)^{n-1}$ as features, along with their labels $(A)^{n-1}$ and $(1-A)^{n-1}$, resulting in a training set of size $2(n-1)$. $\bar{q}_n$ is used to predict $A_n$ based on $\tilde{\mathbf{O}}_n$, and this prediction is compared with the true label $A_n$, as described in equation 5, to update the statistic $W_{n-1}$ to $W_n$ (Here, $\mathbf{S}$ in equation 5 is expressed as $\tilde{\mathbf{O}}$). Moreover, starting from $\lambda_1 = 1$, the betting fraction $\lambda_n$ is computed using ONS algorithm (see Definition 5 in Shekhar & Ramdas (2023)). Finally, $k_n$ returns 1 indicating the rejection of $H_0$ if $W_n \geq \frac{1}{\alpha}$ or otherwise 0.

### B.2 Appropriate Baselines to Consider

As illustrated in Section 3, MPED-RobustCAL is designed to (1) maintain statistical validity, i.e., ensure that the Type I error is bounded above by the pre-selected significance level $\alpha$, (2) achieve higher testing power than conventional MPED, and (3) include more true responders from $\Omega_\gamma$ than existing active designs.

To justify (1), we implemented the practical MPED-RobustCAL on synthetic data generated under $H_0$ and compared the empirical probability of rejecting $H_0$ with the significance level $\alpha$.

To justify (2), we applied the practical MPED-RobustCAL to two real datasets (Atassi et al., 2014; Shalit et al., 2017) under $H_1$ and compared its testing power against the conventional MPED. Several active designs (Li et al., 2024; 2022) aim to maximize testing power by sampling the most informative data points. However, such approaches can lead to the misleading conclusion that the treatment effect is confined to only these highly informative samples. In contrast, MPED-RobustCAL is designed to sample all true responders within a pre-defined target region $\Omega_\gamma$, thereby mitigating the risk of suggesting that the treatment is not broadly applicable to patients. Consequently, while the testing power of MPED-RobustCAL exceeds that of conventional MPED, it remains lower than that of methods that exclusively sample the most informative data points. This reduction in power represents the trade-off that MPED-RobustCAL accepts in order to achieve its third objective: enrolling all true responders in $\Omega_\gamma$.

To justify (3), we compare MPED-RobustCAL with two existing active designs described in Simon & Simon (2013) and Jesson et al. (2021). The active design proposed in Simon & Simon (2013) approximates the noise-free control response $f(\mathbf{x})$ and treatment response $f(\mathbf{x}) + \Delta(\mathbf{x})$ using two regression functions $f_c(\mathbf{x})$ and $f_t(\mathbf{x})$. The enrollment region for this design is then defined as

$$\{\mathbf{x} \in \mathcal{X} \mid f_t(\mathbf{x}) - f_c(\mathbf{x}) \geq \gamma\},$$

where $\gamma$ is the treatment-effect threshold used to define the target $\Omega_\gamma$ in MPED-RobustCAL. We refer to this approach as the *regression-based active design* in this work.

In addition, Jesson et al. (2021) proposed an approach based on Bayesian active learning by disagreement (BALD), termed $\tau$-*BALD*, which actively labels samples to approximate the effect size $\Delta(\mathbf{x})$ in equation 1. $\tau$-BALD constructs a regression function $g(\mathbf{x})$ to model $Y^1(\mathbf{x}) - Y^0(\mathbf{x})$. Since MPED is an experimental design that has access to both $Y^0$ and $Y^1$ for a pair of units $(\mathbf{x}, \mathbf{x}')$, $\tau$-BALD is applicable to modeling the conditional average effect size $\Delta(\mathbf{x})$. The authors of Jesson et al. (2021) apply Bayesian active learning to efficiently learn $g(\mathbf{x})$. Consequently, the enrollment region for this design is defined as

$$\{\mathbf{x} \in \mathcal{X} \mid g(\mathbf{x}) \geq \gamma\}.$$

### B.3    Experiments with synthetic data

#### B.3.1    Data model

*We first describe the simulation data generated under $H_1$.* We simulate a population of participants using a two-dimensional random variable $\mathbf{X} = (X_1, X_2)$, where both $X_1$ and $X_2$ follow uniform distributions between 0 and 1, i.e., $p_{X_1}(x)$ and $p_{X_2}(x)$ are uniform $(0, 1)$. We define

$$f(\mathbf{X}) = X_1 + 2X_1 - X_1 X_2, \quad (X_1, X_2) \sim p_{X_1}(x_1) p_{X_2}(x_2) \tag{6}$$

$$\Delta(\mathbf{X}) = \begin{cases} 1, & \text{if } X_1 + s < X_2 \\ 0, & \text{otherwise} \end{cases} \tag{7}$$

$$E \sim \mathcal{N}(0, \sigma^2). \tag{8}$$

Here, the constant $s$ is *inversely* proportional to the size of high treatment-effect region, while $\sigma^2$ represents the variance of the noise added to experimental outcomes. *For the simulation under $H_1$ in the following*, we set $s = 0.5$ and $\sigma^2 = 0.1$. An illustration of the simulated participants' covariates is provided in Figure 4. Points classified as 1 lie in the high treatment-effect region, defined as $\{\forall \mathbf{x} \in \mathcal{X} \mid \Delta(\mathbf{x}) = 1\}$, while points classified as 0 lie in the zero treatment-effect region, defined as $\{\forall \mathbf{x} \in \mathcal{X} \mid \Delta(\mathbf{x}) = 0\}$.

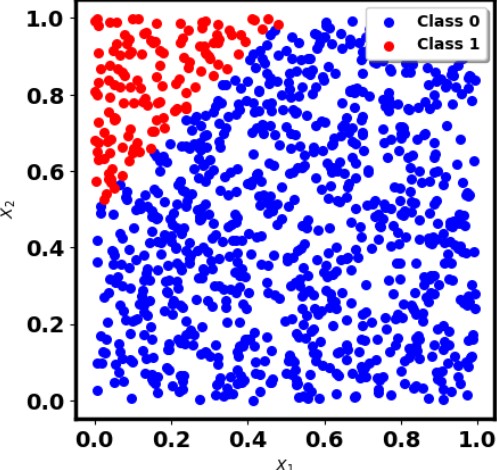

Figure 4: Illustration of the synthetic data. Class 0 and 1 represents points in zero treatment-effect and high treatment-effect regions respectively.

*For the simulation data generated under $H_0$,* we replace equation 7 with $\forall \mathbf{X} \in \mathcal{X}, \Delta(\mathbf{X}) = 0$ indicating that the treatment effect is zero everywhere.

#### B.3.2    Implementation Details

We sample $M = 1000$ data points $(\mathbf{X})^M$ from $p_{X_1 X_2}$. If a participant $\tilde{\mathbf{X}} \in (\mathbf{X})^M$ is selected by practical *MPED-RobustCAL* to be enrolled in the experiment, additional points are sampled from $p_{X_1 X_2}$ until a match $\tilde{\mathbf{X}}'$ is identified such that $\tilde{\mathbf{X}}$ and $\tilde{\mathbf{X}}'$ are sufficiently close. Specifically, we pair

$\tilde{\mathbf{X}}'$ with $\tilde{\mathbf{X}}$ when $||\tilde{\mathbf{X}} - \tilde{\mathbf{X}}'||_2 \leq 0.01$. As noted in Balzer et al. (2012) and van der Laan et al. (2012), it is conventional to consider sampling $\tilde{\mathbf{X}}'$ from a distribution conditional on the value of $\tilde{\mathbf{X}}$ in order to form the matched pair $\left( \tilde{\mathbf{X}}, \tilde{\mathbf{X}}' \right)$.

We implement the practical *MPED-RobustCAL* in Algorithm 2 to actively enroll participants from $\mathcal{S} = (\mathbf{X})^M$. We define a classifier set $\mathcal{C}$ consisting of 10 logistic regression models and initialize the training set $Q$ with 50 randomly labeled data points queried from $\mathcal{S}$. Specifically, we generate 10 different training sets by bootstrapping $Q$, and train each classifier in $\mathcal{C}$ using one of these sets. As more labeled data is added to $Q$ during the algorithm's execution, the same procedure is used to update $\mathcal{C}$. We set the significance level $\alpha = 0.05$, the treatment effect threshold $\gamma = 0.2$, and evaluate the performance of *MPED-RobustCAL* across various label budgets $B$, ranging from 200 to 700 in increments of 100. We use the sequential predictive two-sample test (Podkopaev & Ramdas, 2023) to instantiate $k$, as outlined in Algorithm 3. Additionally, a logistic regression classifier $\bar{q}$ is employed in Algorithm 3 to perform the two-sample test.

We run the simulation 100 times, with simulation data randomly generated for each iteration, and compare the performance of the conventional MPED and *MPED-RobustCAL*, summarizing the results across the 100 simulations.

### B.3.3 TESTING POWER AND STOPPING TIME UNDER $H_1$

Table 3 presents the testing power of the conventional MPED and *MPED-RobustCAL*. As shown, *MPED-RobustCAL* achieves a higher probability of correctly rejecting $H_0$ (i.e., higher testing power) across 100 simulations. This improvement can be attributed to *MPED-RobustCAL*'s ability to actively and adaptively identify an enrollment region with a high treatment effect, selectively enrolling participants from this region in the experiments. In contrast, the conventional MPED randomly enrolls participants from the entire population, causing a significant portion of the labeling or experimental budget to be spent on zero treatment-effect regions.

Table 3: A comparison of the *testing power* between the conventional MPED and the proposed *MPED-RobustCAL* across label budgets.

| Label budget | 200 | 300 | 400 | 500 | 600 | 700 |
|---|---|---|---|---|---|---|
| Conventional | 0.07 | 0.11 | 0.15 | 0.18 | 0.19 | 0.22 |
| *MPED-RobustCAL* | **0.16** | **0.34** | **0.61** | **0.76** | **0.85** | **0.85** |

Table 4: A comparison of the *average stopping time* between the conventional MPED and the proposed *MPED-RobustCAL* across various label budgets.

| Label budget | 200 | 300 | 400 | 500 | 600 | 700 |
|---|---|---|---|---|---|---|
| Conventional | 193.63±26.09 | 285.89±50.45 | 375.35±77.03 | 460.71±106.57 | 542.69±138.60 | 620.75±172.70 |
| *MPED-RobustCAL* | **157.93**±47.30 | **179.61**±71.48 | **181.32**±75.07 | **182.06**±77.56 | **182.06**±77.56 | **182.06**±77.56 |

In addition to testing power, another important evaluation metric is stopping time, which refers to the number of labels required to reject $H_0$ within a given label budget. Experimental designs that consistently select participants from high treatment-effect regions tend to use fewer labels compared to designs that allocate a significant portion of the budget to zero treatment-effect regions. Table 4 shows that the average number of labels used by *MPED-RobustCAL* across various label budgets is consistently smaller than that of the conventional MPED.

### B.3.4 ENROLLMENT REGION UNDER $H_1$

We highlight the differences in the participants selected by the conventional MPED and *MPED-RobustCAL* in Figure 5. As expected, *MPED-RobustCAL* actively enrolls participants covering a region which encloses the high treatment-effect area highlighted in red. Additionally, this enrollment region is smaller than the entire population space, leading to improved testing power as shown in Table 1.

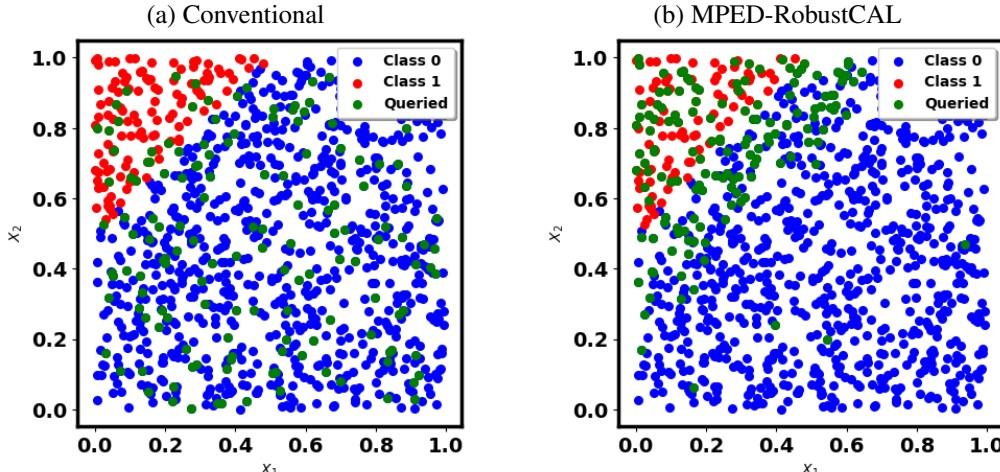

Figure 5: Illustration of the labeled data obtained by the conventional MPED and *MPED-RobustCAL*. Classes 0 and 1 represent points in zero and high treatment-effect regions, respectively. Participants randomly selected to initialize the classifiers in *MPED-RobustCAL* are excluded.

In addition to the visualization of labeled points, we also calculate the true positive rates (TPR) and precision of *MPED-RobustCAL* along with the increasing label budget, using a validation set of simulation data. TPR represents the ratio of *true* high treatment-effect points enrolled by *MPED-RobustCAL* to the total high treatment-effect points (i.e., points highlighted by red in Figure 4). Precision, on the other hand, represents the ratio of *true* high treatment-effect points enrolled by *MPED-RobustCAL* to all points enrolled by *MPED-RobustCAL*. Both metrics are calculated using the validation set, which is only used to demonstrate our theoretical analysis in Theorem 4.5. This validation set is not required in the practical implementation of *MPED-RobustCAL* in real-world experiments. As noted in Theorem 4.5, the enrollment region identified by *MPED-RobustCAL* is a superset of the target region, and this superset reduces faster than passive learning.

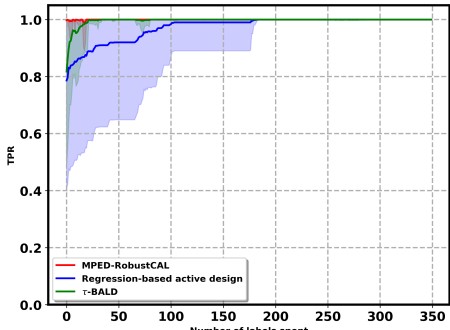

Figure 6: A comparison of the TPR between *MPED-RobustCAL* and the active design in Simon & Simon (2013).

To demonstrate Theorem 4.5, which states that *MPED-RobustCAL* identifies an enrollment region enclosing the target region, we compare the TPR of *MPED-RobustCAL* with the regression-based active design in Simon & Simon (2013) and $\tau$-BALD in Jesson et al. (2021). Gaussian process regressions are employed to construct the regression functions for both approaches. Figure 6 shows that *MPED-RobustCAL* maintains a higher TPR along with the label budget compared to these two existing active designs in Simon & Simon (2013) and Jesson et al. (2021), indicating that most points in the target region are included in the enrollment region identified by *MPED-RobustCAL*, as suggested by Theorem 4.5.

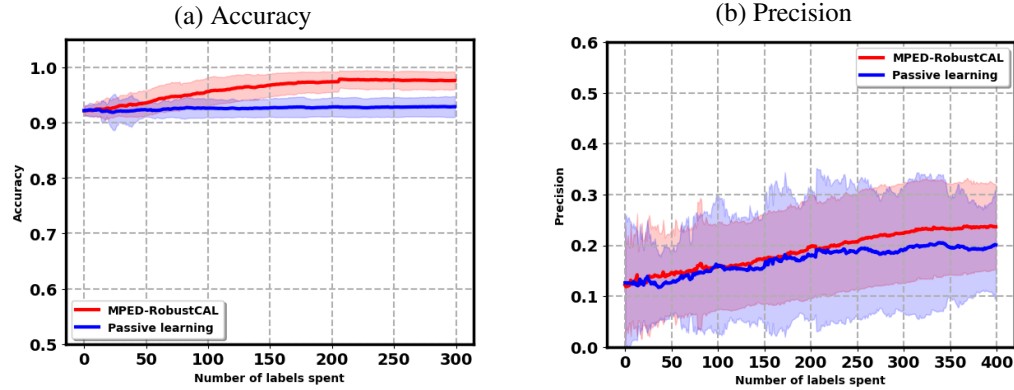

Figure 7: Accuracy and precision by *MPED-RobustCAL* and passive learning.

Finally, we provide a comparison of precision and accuracy between *MPED-RobustCAL* and passive learning in Figure 7. As observed, both the precision and accuracy achieved by *MPED-RobustCAL* increase more rapidly than those achieved by passive learning, corroborating Theorem 4.5. This theorem states that *MPED-RobustCAL* requires fewer labels than passive learning to attain the same ratio of the enrollment region size to the target region size, as well as the same classifier error rate.

### B.3.5   TYPE I ERROR UNDER $H_0$

Theorem 4.5 implies that the Type I error of *MPED-RobustCAL* is still upper-bounded by $\alpha$, even the participants are actively enrolled. Figure 8 demonstrate this, showing that the Type I errors of *MPED-RobustCAL* are all smaller than $\alpha = 0.05$ on various label budget.

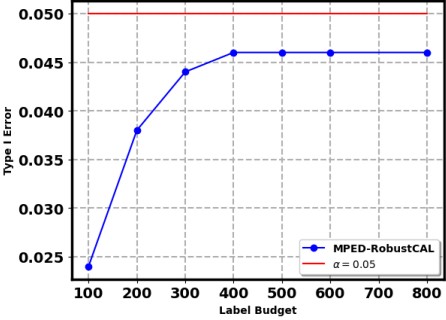

Figure 8: Type I error by *MPED-RobustCAL* across various label budgets

### B.4   EXPERIMENTS WITH AMYOTROPHIC LATERAL SCLEROSIS DATA

This section presents experimental results for implementing *MPED-RobustCAL* (Algorithm 2) using PRO-ACT, an Amyotrophic Lateral Sclerosis (ALS) dataset described in Atassi et al. (2014).

### B.4.1   DATA DESCRIPTION

ALS is a neurological disease that causes progressive muscle weakness and can ultimately lead to paralysis. Pharmaceutical scientists develop prototype medication treatments and design clinical trials to validate their effectiveness. Specifically, the Pooled Resource Open-Access ALS Clinical Trials database (PRO-ACT), as detailed in (Atassi et al., 2014), provides experimental outcomes from patients who received Riluzole, a drug already approved by the U.S. Food and Drug Administration (FDA). In conventional clinical trials, the MPED randomly enrolls participants, which can lead to inefficient use of experimental resources. To address this, we apply the proposed *MPED-RobustCAL* to actively enroll participants in regions with high treatment effects, thereby reducing the experimental

budget required to determine the effectiveness of Riluzole. The PRO-ACT database provides the ALS Functional Rating Scale (ALSFRS), which includes 10 assessments of ALS patients' motor function. From these, we selected scores for "Speech", "Salivation", "Swallowing", "Handwriting", "Cutting food and handling utensils", "Dressing and hygiene", "Walking", "Climbing stairs", and "Breathing" to construct a dataset of matched pairs, where each participant is represented by nine covariates. This data creation process is repeated by resampling participants from the entire PRO-ACT dataset, resulting in 100 datasets, each containing around 770 matched-pairs. Furthermore, the experimental outcome is defined as the slope of the ALSFRS, which indicates the decline of the sum of nine assessment scores over a (roughly) similar duration of time. A smaller slope in the treatment group compared to the control group indicates that the treatment (i.e., Riluzole) is effective in slowing the decline of ALSFRS scores in ALS patients.

### B.4.2 IMPLEMENTATION DETAILS

We employ practical *MPED-RobustCAL* by setting the treatment-effect threshold $\gamma = 0.1$, significance level $\alpha = 0.05$ and the label budget $B$ ranging from 250 to 400 in increments of 50. To evaluate the sensitivity of enrollment region identification to the choice of classifier, we utilize three sets of classifiers: logistic regression, k-nearest neighbors (KNN), and decision tree. Finally, we use a separate decision tree for the instantiation of the sequential test $k$, implemented through the sequential predictive test.

### B.4.3 TESTING POWER AND STOPPING TIME UNDER $H_1$

As the treatment, Riluzole, is a drug approved by FDA, demonstrating that it is an effective medication for ALS, the experimental data is considered to be generated under $H_1$. Table 5 compares the testing power of *MPED-RobustCAL* and the conventional MPED across various classifier sets. As observed,

Table 5: A comparison of the *testing power* between the conventional MPED and the proposed *MPED-RobustCAL* across label budgets, using various classifier sets.

(a) Logistic Regression

| Label | 250 | 300 | 350 | 400 |
|---|---|---|---|---|
| Conventional | 0.10 | 0.29 | 0.40 | 0.67 |
| *MPED-RobustCAL* | **0.18** | **0.36** | **0.61** | **0.81** |

(b) Decision Tree

| Label | 250 | 300 | 350 | 400 |
|---|---|---|---|---|
| Conventional | 0.10 | 0.29 | 0.40 | 0.67 |
| *MPED-RobustCAL* | **0.19** | **0.39** | **0.59** | **0.79** |

(c) k-NN

| Label | 250 | 300 | 350 | 400 |
|---|---|---|---|---|
| Conventional | 0.10 | 0.29 | 0.40 | 0.67 |
| *MPED-RobustCAL* | **0.12** | **0.30** | **0.56** | **0.73** |

*MPED-RobustCAL* effectively improves the testing power compared to the conventional MPED across all three classifier sets. It is worth noting that the testing powers for the conventional MPED in Table 5 (a), (b), and (c) remain identical, as the conventional MPED randomly enrolls participants from the original population regardless of the classifier used. In addition to testing power, we also evaluate the number of labels spent, or the stopping time, within each budget. These results are presented in Table 6. As observed, *MPED-RobustCAL* achieves a smaller stopping time compared to the conventional MPED in each comparison.

Table 6: A comparison of the *average stopping time* between the conventional MPED and the proposed *MPED-RobustCAL* across label budgets, using various classifier sets.

(a) Logistic regression

| Label budget | 250 | 300 | 350 | 400 |
|---|---|---|---|---|
| Conventional | 244.45±24.02 | 285.75±36.30 | 318.29±52.68 | 341.24±68.81 |
| *MPED-RobustCAL* | **236.41**±39.67 | **273.02**±54.60 | **299.81**±70.37 | **313.58**±82.04 |

(b) Decision tree

| Label budget | 250 | 300 | 350 | 400 |
|---|---|---|---|---|
| Conventional | 244.45±24.02 | 285.75±36.30 | 318.29±52.68 | 341.24±68.81 |
| *MPED-RobustCAL* | **236.24**±42.10 | **271.48**±56.35 | **298.69**±72.53 | **313.30**±84.72 |

(c) k-NN

| Label budget | 250 | 300 | 350 | 400 |
|---|---|---|---|---|
| Conventional | 244.45±24.02 | 285.75±36.30 | 318.29±52.68 | 341.24±68.81 |
| *MPED-RobustCAL* | **238.81**±41.15 | **278.46**±53.97 | **306.24**±67.97 | **323.28**±81.18 |

### B.4.4 Rate of True positive regions

Theorem 4.5 in the main paper suggests that *MPED-RobustCAL* is an experimental design capable of ensuring that the enrollment region covers the target region $\Omega_\gamma$, which corresponds to points with high treatment effects. To evaluate this, we present the results for the true positive rate (TPR), defined as the ratio of the number of points $\mathbf{x}$ with labels 1 (i.e., points with high treatment effects) included in the enrollment region to the total number of points with labels 1.

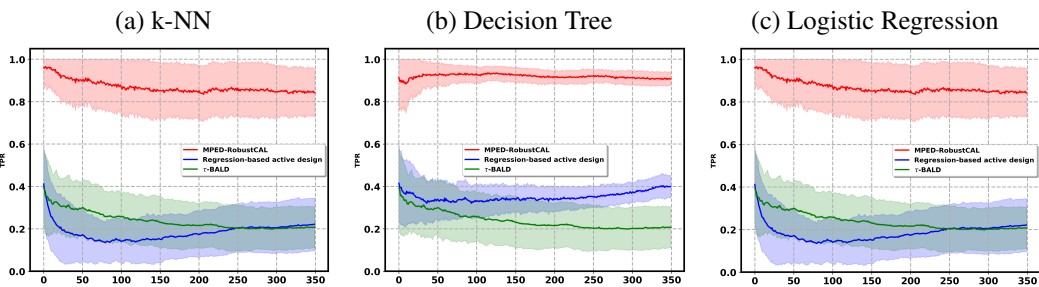

(a) k-NN    (b) Decision Tree    (c) Logistic Regression

Figure 9: TPR comparison between *MPED-RobustCAL* and a standard active design across various label budgets.

Additionally, we compare the TPR of *MPED-RobustCAL* with the regression-based active design proposed in Simon & Simon (2013) and with $\tau$-BALD from Jesson et al. (2021). For the regression-based active design, $k$-nearest neighbors (kNN), decision trees, and logistic regression are used to construct the regression functions, while for $\tau$-BALD a Gaussian process is employed to construct the regression function. The TPRs are computed by identifying enrollment points from an independent validation set under various label budgets. The use of this validation set allows for an unbias TPR comparison among *MPED-RobustCAL* and the designs in Simon & Simon (2013) and (Jesson et al., 2021). However, it is important to note that this validation set is only used for the TPR comparison and is not required for the practical implementation of *MPED-RobustCAL*. The comparative results are presented in Figure 9, showing the average TPR calculated from 100 validation sets sampled from the entire ALS dataset. Three classifier sets, including knn, decision tree and logistic regression, are employed to construct the classifier committee. As observed, *MPED-RobustCAL* consistently achieves a significantly higher TPR compared to the two active design baselines across various label budgets, as expected. Ideally, the TPR for *MPED-RobustCAL* converges to one, as demonstrated in the results of the synthetic data presented in Figure 6. However, the labels of points in the PRO-ACT dataset contain noise, meaning that points labeled as one—based on the comparison of treatment and control responses—do not always accurately indicate that the points belong to $\Omega_\gamma$. Furthermore, unlike the synthetic data, we do not have perfect identification of points in $\Omega_\gamma$ for the PRO-ACT dataset. This lack of perfect ground-truth of $\Omega_\gamma$ leads to the TPR for *MPED-RobustCAL* not converging to one.

### B.5 Experiments with the Infant Health and Development Program dataset

#### B.5.1 Data description

The Infant Health and Development Program (IHDP) dataset (Shalit et al., 2017) contains data for studying the effect of home visits by specialist doctors on the cognitive test scores of premature infants. In this dataset, the treatment/control assignment $A$ indicates whether a participant received a home visit, and the outcome represents the cognitive test score. The dataset includes approximately 750 subjects and 25 covariates. Specifically, both factual and counterfactual outcomes are available for each participant. Therefore, we perform simulations under an exact-match setting, i.e., $\tilde{\mathbf{X}} = \tilde{\mathbf{X}}'$ for each pair $\left(\tilde{\mathbf{X}}, \tilde{\mathbf{X}}'\right)$, using the IHDP dataset.

### B.5.2 IMPLEMENTATION DETAILS

We implement Algorithm 2 to actively enroll participants from $\mathcal{S} = (\mathbf{X})^M$. The testing function $k$ is instantiated using the sequential predictive test proposed in Podkopaev & Ramdas (2023), as presented in Algorithm 3. The labeled set $Q$ is bootstrapped to generate 10 training subsets, which are used to initialize or update an ensemble $\mathcal{C}$ consisting of 10 decision tree, or k-NN models, respectively. The training set $Q$ is initialized with 10 randomly labeled data points from the IHDP dataset. The significance level is set to $\alpha = 0.05$, and the treatment effect threshold $\gamma$ is set to 4.5. *The simulation on IHDP is conducted solely to evaluate the performance of identifying the target region* $\Omega_\gamma$. This is because the average treatment effect across the entire IHDP population is already high, and random enrollment alone yields high testing power. We perform sampling using the entire IHDP dataset, generating 100 subsets, each containing approximately 530 subjects.

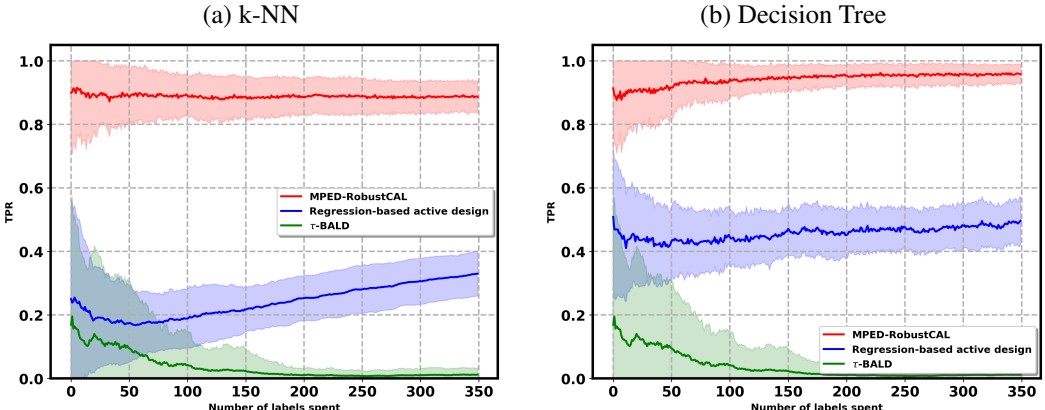

Figure 10: TPR comparison between *MPED-RobustCAL* and a standard active design across various label budgets.

### B.5.3 RATE OF TRUE POSITIVE REGIONS

Similar to Section B.4.4, we compare the TPR of *MPED-RobustCAL* with the regression-based active design proposed in Simon & Simon (2013) and with $\tau$-BALD from Jesson et al. (2021). For the regression-based active design, $k$-nearest neighbors (kNN) and decision trees are used to construct the regression functions, while for $\tau$-BALD a Gaussian process is employed to construct the regression function. The TPRs are computed by identifying enrollment points from an independent validation set under various label budgets. The use of this validation set allows for an unbias TPR comparison among *MPED-RobustCAL* and the designs in Simon & Simon (2013) and Jesson et al. (2021). The comparative results between *MPED-RobustCAL* and the active designs in Simon & Simon (2013) and Jesson et al. (2021) are presented in Figure 9, showing the average TPR computed over 100 validation sets sampled from the entire IHDP dataset. Two types of classifier sets—k-NN and decision tree—are employed to construct the classifier committee. As observed, *MPED-RobustCAL* consistently achieves a significantly higher TPR than the active designs in Simon & Simon (2013) and Jesson et al. (2021) across various label budgets, as expected.

### B.5.4 EVALUATIONS OF THE PRECISION

Theorem 4.5 suggests that the enrollment region converges to the target region $\Omega_\gamma$ more rapidly under *MPED-RobustCAL* than with passive learning. This implies that the true positive rate (TPR) for both approaches remains close to one throughout the classifier's training, indicating that most responders within the target region are eventually retrieved. However, the precision achieved by active learning improves at a faster rate than that of passive learning. As noted in Remark 4.6 in the main paper, the learning efficiency of *MPED-RobustCAL* is lower than that of the original RobustCAL, due to the additional label queries allocated to the positive region POS $(\mathcal{C})$ to facilitate two-sample testing. Figure 11 supports both Theorem 4.5 and Remark 4.6, showing that while both active and passive

learning achieve high TPR, the precision under active learning increases at a moderately faster rate or remains comparable.

(a) k-NN                                      (b) Decision Tree

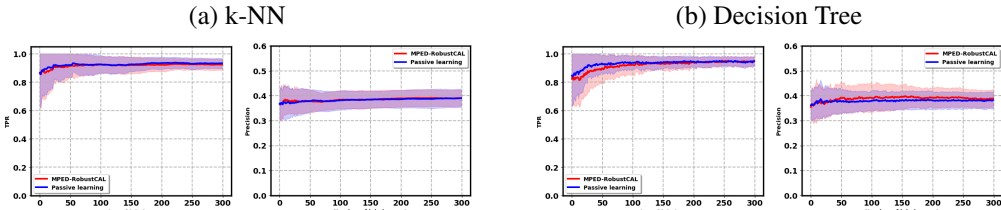

Figure 11: Comparison of TPR and precision between passive learning and *MPED-RobustCAL* across various label budgets. Each subfigure shows TPR (left) and precision (right) for a given classifier.

## B.6    SENSITIVITY ANALYSIS ON THE HYPERPARAMETERS OF PRACTICAL MPED-ROBUSTCAL

The number of labeled samples used to initialize the committee of classifiers and the size of the committee are two main hyperparameters for Practical MPED-RobustCAL in Algorithm 2. In this section, we evaluate the sensitivity of these hyperparameters by examining the testing power under different settings. Table 7 presents a comparison of testing power, evaluated using the PRO-ACT dataset, between Practical RobustCAL and conventional MPED as the number of initial labeled samples and the committee size vary. Decision trees are used to construct the classifiers. The results show that, across most settings, MPED-RobustCAL achieves higher testing power than conventional MPED.

Table 7: Testing power comparison by committee size and initial label size. Bold indicates MPED-RobustCAL outperforming Conventional MPED.

| Initial Label Size = 10 | | | | | |
|---|---|---|---|---|---|
| Method | # Classifiers | 250 | 300 | 350 | 400 |
| Conventional MPED | – | 0.14 | 0.27 | 0.47 | 0.70 |
| MPED-RobustCAL | 2 | **0.17** | **0.32** | **0.55** | **0.78** |
| MPED-RobustCAL | 4 | 0.09 | 0.27 | **0.48** | **0.80** |
| MPED-RobustCAL | 6 | **0.16** | **0.34** | **0.58** | **0.81** |
| MPED-RobustCAL | 8 | **0.22** | **0.35** | **0.57** | **0.76** |
| MPED-RobustCAL | 10 | **0.19** | **0.39** | **0.59** | **0.79** |

| Initial Label Size = 30 | | | | | |
|---|---|---|---|---|---|
| Method | # Classifiers | 250 | 300 | 350 | 400 |
| Conventional MPED | – | 0.12 | 0.26 | 0.49 | 0.70 |
| MPED-RobustCAL | 2 | **0.13** | **0.29** | **0.56** | **0.75** |
| MPED-RobustCAL | 4 | **0.13** | **0.30** | **0.51** | 0.69 |
| MPED-RobustCAL | 6 | **0.16** | **0.30** | **0.54** | 0.70 |
| MPED-RobustCAL | 8 | **0.18** | **0.27** | **0.55** | **0.77** |
| MPED-RobustCAL | 10 | **0.13** | **0.31** | 0.47 | **0.77** |

## B.7    PERFORMANCE OF PRACTICAL MPED-ROBUSTCAL UNDER VARIOUS PROBLEM DIFFICULTIES

The difficulty of the synthetic dataset is controlled by adjusting the intercept value from 0.6 to 0, where a smaller intercept corresponds to an easier problem (see Appendix B.3.1 for details of the data model). Accordingly, $P(\Omega_\gamma)$—a probabilistic upper bound on POS $(C)$—increases as the problem becomes easier. We report simulation results on testing power, comparing MPED-RobustCAL with

conventional MPED across different difficulty levels in the synthetic dataset. Table 8 shows that when the problem is easy, the testing power of conventional MPED is already high. The advantage of MPED-RobustCAL becomes more pronounced as the problem increases in difficulty (e.g., at intercept = 0.2).

Table 8: Testing power for various intercepts used to generate synthetic data (label budget = 200). Larger intercepts represent harder problems, corresponding to smaller $\Omega_\gamma$.

| Intercept | 0.0 | 0.1 | 0.2 | 0.3 | 0.4 | 0.5 |
|---|---|---|---|---|---|---|
| Conventional MPED | 1.00 | 0.95 | 0.72 | 0.41 | 0.18 | 0.06 |
| MPED-RobustCAL | 1.00 | **1.00** | **0.99** | **0.96** | **0.73** | **0.23** |

## C    PROOF OF PROPOSITION 4.1

*Proof.* We divide $\mathcal{X}$, the support of $p_{\mathbf{X}}$, into two regions: $\Omega_\gamma = \{\mathbf{x} \in \mathcal{X} \mid \Delta(\mathbf{x}) \geq \gamma\}$ and $\Omega_{\bar{\gamma}} = \{\mathbf{x} \in \mathcal{X} \mid \Delta(\mathbf{x}) < \gamma\}$. Assumption 3.2 states that the observed control and treatment *r.v.* $(Y^0, Y^1)$ are independent of the treatment assignment $A$ conditional on $(\mathbf{X}, \mathbf{X}')$. This implies that,

$$\forall (\mathbf{X}, \mathbf{X}') \in \mathcal{X} \times \mathcal{X}, Y^0(\mathbf{X}) = Y^0(\mathbf{X}') \text{ and } Y^1(\mathbf{X}) = Y^1(\mathbf{X}'). \tag{9}$$

Consequently, we define $G(\mathbf{X}) = Y^1(\mathbf{X}) - Y^0(\mathbf{X})$ to represent the *r.v.* indicating the outcome difference between treatment and control experiments within a matched-pair. *MPED-RobustCAL* assigns the label $\tilde{Z} = 1$ to $\mathbf{x}$ if $G(\mathbf{x}) \geq \gamma$, and $\tilde{Z} = 0$ otherwise. Therefore, $P_{\tilde{Z}|\mathbf{X}}\left(\tilde{Z} = 1 \mid \mathbf{X}\right) = P(G(\mathbf{X}) - \gamma \geq 0 \mid \mathbf{X})$ and $P_{\tilde{Z}|\mathbf{X}}\left(\tilde{Z} = 0 \mid \mathbf{X}\right) = P(G(\mathbf{X}) - \gamma < 0 \mid \mathbf{X})$. From the data model in equation 1, where $Y^A$ contains zero-mean noise $E \sim \mathcal{N}(0, \sigma^2)$, we have:

$$G(\mathbf{x}) \sim \mathcal{N}\left(\mu_\gamma(\mathbf{x}), \sigma^2\right), \mu_\gamma(\mathbf{x}) \geq \gamma, \quad \forall \mathbf{x} \in \Omega_\gamma \tag{10}$$

$$G(\mathbf{x}) \sim \mathcal{N}\left(\mu_{\bar{\gamma}}(\mathbf{x}), \sigma^2\right), \mu_{\bar{\gamma}}(\mathbf{x}) < \gamma, \quad \forall \mathbf{x} \in \Omega_{\bar{\gamma}} \tag{11}$$

Consequently,

$$P_{\tilde{Z}|\mathbf{X}}\left(\tilde{Z} = 1 \mid \mathbf{X}\right) = P(G(\mathbf{X}) - \gamma \geq 0 \mid \mathbf{X}) \geq 0.5, \quad \forall \mathbf{X} \in \Omega_\gamma \tag{12}$$

$$P_{\tilde{Z}|\mathbf{X}}\left(\tilde{Z} = 0 \mid \mathbf{X}\right) = P(G(\mathbf{X}) - \gamma < 0 \mid \mathbf{X}) < 0.5, \quad \forall \mathbf{X} \in \Omega_{\bar{\gamma}} \tag{13}$$

Hence, the Bayes optimal classifier $q^*(\mathbf{x}) = \begin{cases} 1 \text{ if } P_{\tilde{Z}|\mathbf{X}}(1 \mid \mathbf{x}) \geq 0.5 \\ 0 \text{ otherwise} \end{cases}$, assigns 1 to $\forall \mathbf{x} \in \Omega_\gamma$ and 0 to $\forall \mathbf{x} \in \Omega_{\bar{\gamma}}$. $\qquad\square$

## D    PROOF OF THEOREM 4.5

Our proof of Theorem 4.5 closely resembles the proof of Theorem 5.4 in Hanneke et al. (2014) for the original RobustCAL algorithm. However, our proof has been adapted to accommodate the proposed *MPED-RobustCAL*. For details on the original proof for RobustCAL, we refer readers to Section 5.2 of Hanneke et al. (2014).

*Proof.* The proof of Theorem 4.5 is established under the following assumption with respect to $p_{\mathbf{X}\tilde{Z}}$,

**Assumption D.1.** *(Tsybakov (2004)) Given $p_{\mathbf{X}\tilde{Z}}$, a classifier set $\mathbb{C}$ and a Bayes optimal classifier $q^*(\mathbf{x})$ with respect to $p_{\mathbf{X}\tilde{Z}}$, there exist constants $a \in [1, \infty)$ and $\rho \in [0, 1]$ such that, for every $h \in \mathbb{C}$, the following holds*

$$P(h(\mathbf{X}) \neq q^*(\mathbf{X})) \leq a(er(h) - er(q^*))^\rho \tag{14}$$

*where $er(h)$ represents the classification error of $h$ over $p_{\mathbf{X}\tilde{Z}}$.*

The authors of Massart & Nedelec (2006) establish that a bounded noise condition implies Assumption D.1 for $\rho = 1$,

**Assumption D.2.** *(Bounded noise condition (Massart & Nedelec, 2006)) Given $\tilde{\eta}(\mathbf{x}) = P_{\tilde{Z}|\mathbf{X}}\left(\tilde{Z} = 1 \mid \mathbf{X}\right)$ with respect to $p_{\mathbf{X}\tilde{Z}}$, there exists $a \in [1, \infty)$ such that*

$$P\left(\mathbf{X} : |\tilde{\eta}(\mathbf{X}) - 1/2| < 1/(2a)\right) = 0 \tag{15}$$

*where $\mathbf{X} \sim p_{\mathbf{X}}$.*

Assumption D.2 is stated earlier in Assumption 4.4, indicating that $\tilde{\eta}(\mathbf{x})$ is bounded away from $1/2, \forall \mathbf{x} \in \mathcal{X}$. Additionally, Assumption D.2 implies that $\tilde{\eta}(\mathbf{x}) \neq 1/2, \forall \mathbf{x} \in \mathcal{X}$, which addresses scenarios under $H_1$. Under $H_0$, $\mathbf{X}$ and $\tilde{Z}$ are independent, making the classification problem trivial. Consequently, an adapted version of Assumption D.2 relevant to our work is presented in Assumption 4.4. We restate it here for the reader's convenience.

**Assumption D.3.** *(Bounded noise condition (Massart & Nedelec, 2006)) Under $H_1$, there exists $a \in [1, \infty)$ such that*

$$P\left(\mathbf{X} : |\tilde{\eta}(\mathbf{X}) - 1/2| < 1/(2a)\right) = 0 \tag{16}$$

*where $\mathbf{X} \sim p_{\mathbf{X}}$, and furthermore, the Bayes optimal classifier $q^* \in \mathbb{C}$.*

This adapted Assumption D.3 further assumes the Bayes optimal classifier $q^* \in \mathbb{C}$. Herein, we restated the definition of *Vapnik-Chervonenkis* (VC) dimension of a classifier class $\mathbb{C}$.

**Definition D.4.** *(VC dimension (Vapnik & Chervonenkis, 2015)) The VC dimension of a non-empty $\mathbb{C}$ is the largest integer $m$ such that there exists a set of $m$ points, $(\mathbf{x})^m$, and for any label assignments to the points in $(\mathbf{x})^m$, there always exists $h \in \mathbb{C}$ that can perfectly classify them.*

An important lemma, which will be used throughout the proof, is stated in the following,

**Lemma D.5.** *(Concentration inequalities (Hanneke et al., 2014)) Given $p_{\mathbf{X}\tilde{Z}}$, a classifier set $\mathbb{C}$ and a Bayes optimal classifier $q^*$ with respect to $p_{\mathbf{X}\tilde{Z}}$, there is a universal constant $c \in [1, \infty)$ such that, for $\left(\mathbf{X}, \tilde{Z}\right)^m$ i.i.d. sampled from $p_{\mathbf{X}\tilde{Z}}$, the following holds with probability at least $1 - \delta, \forall h \in \mathbb{C}$*

$$er(h) - er(q^*) \leq \max\{2(er_m(h) - er_m(q^*)), \epsilon\} \tag{17}$$

$$er_m(h) - \min_{g \in \mathbb{C}} er_m(g) \leq \max\{2(er(h) - er(q^*)), \epsilon\} \tag{18}$$

*when*

$$m \geq c \ \max \begin{cases} a\epsilon^{\rho-2}\left(d_{vc}\log\left(\theta_{q^*}(a\epsilon^\rho)\right) + \log(1/\delta)\right) \\ \left(\frac{\beta+\epsilon}{\epsilon^2}\right)\left(d_{vc}\log\left(\theta_{q^*}(\beta + \epsilon)\right)\right) + \log(1/\delta)), \end{cases} \tag{19}$$

*where $d_{vc}$ is the VC-dimension of $\mathbb{C}$, $\beta$ is the Bayes error rate of $q*$, and $\theta_{q^*}$ is the disagreement coefficient introduced in Definition 4.3 in the main paper.*

Lemma D.5 originates from the work of Giné & Koltchinskii (2006), which states $\epsilon$ as a function of $m$. Replacing $\epsilon$ in equation 19 with $U(m, \delta)$, the authors of Giné & Koltchinskii (2006) presents that, given a sample complexity $m$, the concentration inequalities in equation 17 and equation 18 hold with probability at least $1 - \delta$ for

$$U(m, \delta) = \hat{c} \min \begin{cases} \left(\frac{a\left(d_{vc}\log\left(\theta_{q^*}\left(a\left(\frac{a d_{vc}}{m}\right)^{1/(2-\rho)}\right)\right) + \log(1/\delta)\right)}{m}\right)^{\frac{1}{2-\rho}} \\ \frac{d_{vc}\log\left(\theta_{q^*}(d_{vc}/m)\right) + \log(1/\delta)}{m} + \sqrt{\frac{\beta\left(d_{vc}\log\left(\theta_{q^*}(\beta)\right)\right) + \log(1/\delta)}{m}} \end{cases} \tag{20}$$

where $\hat{c} \in (1, \infty)$ is an universal constant. Lemma D.5 provides a tool to analyze the sample complexity needed to acquire a classifier with the excess error $\epsilon$ compared to the Bayes classifier $q^*$.

The *passive learning* result is presented in Section 3.3 in Hanneke et al. (2014). We restate their results in the following,

**Theorem D.6.** *Under Assumption D.1, passive learning attains a classifier $h \in \mathbb{C}$ such that $er(h) - er(q^*) \leq \epsilon$ with probability at least $1 - \delta$ for any $p_{\mathbf{X}\tilde{Z}}$, using the label complexity at most:*

$$a \left(\frac{1}{\epsilon}\right)^{2-\rho} \left(d_{vc} \log\left(\theta_{q^*} \left(a\epsilon^\rho\right)\right) + \log\left(1/\delta\right)\right). \tag{21}$$

The following proof is comprised of demonstrating that $q^*$ is included in $\mathcal{C}$ throughout the execution of *MPED-RobustCAL* in Algorithm 1, provided that the concentration inequalities in Lemma D.5 holds, and analyzing the label complexity incurred at the end of the execution. This analysis leads to label complexity needed to achieve a classifier with an excess error $\epsilon$ compared with $q^*$. Furthermore, as presented in this analysis, the ratio $\mathcal{R} = \frac{|\text{DIS}(\mathcal{C}) \bigcup \text{POS}(\mathcal{C})|}{|\Omega_\gamma|}$, which represents the ratio of the enrollment region to the target region, is tied to $\epsilon$.

We write $M \subseteq \{0, \cdots, 2^B\}$ to denote the set of values of $m$ obtained during the execution of *MPED-RobustCAL* in Algorithm 1. We write $\mathcal{C}_m$ and $Q_m$ to denote the sets of classifiers and labeled data when the $m_{\text{th}}$ unlabeled $\tilde{\mathbf{X}}$ is sampled from $p_{\mathbf{X}}$ before entering Line 4 in Algorithm 1. Furthermore, for each $m \in M$ with $\log_2(m) \in \mathbb{N}$, we define $\bar{U}(m, \delta)$ in Algorithm 1 as

$$\bar{U}(m, \delta) = U(m, \delta_m) \tag{22}$$

where $\delta_m = \delta / (\log_2(2m))^2$. The value of $\bar{U}(m, \delta)$ is to ensure the total failure probability of the algorithm sums up to at most $\delta$.

We define $E_0$ as the event that the concentration inequalities in Lemma D.5 hold for every $m \in M$ and $\delta_m$ with $m$ satisfying $\log_2 m \in \mathbb{N}$. Then, by using the union bound, the event $E_0$ holds with at least $1 - \sum_{i=1}^{\infty} \frac{\delta}{(1+i)^2} > 1 - 2\delta/3$, implying that for every $m \in M$ and $\delta_m$ with $\log_2 m \in \mathbb{N}$,

$$er_m(q^*) - \min_{g \in \mathbb{C}} er_m(g) \leq U(m, \delta_m), \tag{23}$$

and additionally,

$$er(h) - er(q^*) \leq \max\{2\left(er_m(h) - er_m(q^*)\right), U(m, \delta_m)\}, \forall h \in \mathbb{C}. \tag{24}$$

Furthermore, as *MPED-RobustCAL* only labels points with which $h, g \in \mathcal{C}_{m-1}$ disagrees, then we have $(er_{Q_m}(h) - er_{Q_m}(g))|Q_m| = (er_m(h) - er_m(g))m, m > 0$. Assuming $q^* \in \mathcal{C}_{m-1}$ for some $m \in M$ and $\delta_m$, then

$$(er_{Q_m}(h) - er_{Q_m}(q^*))|Q_m| = (er_m(h) - er_m(q^*))m, \quad \forall h \in \mathcal{C}_{m-1}. \tag{25}$$

Combining equation 23 and equation 25 leads to

$$\left(er_{Q_m}(q^*) - \min_{g \in \mathcal{C}_{m-1}} er_{Q_m}(g)\right)|Q_m| \leq U(m, \delta_m)m, \tag{26}$$

implying $q^*$ is also included in $\mathcal{C}_m$ in the execution of *MPED-RobustCAL*, given Line 9 in Algorithm 1. Furthermore, as $q^* \in \mathbb{C}$ stated in Assumption D.3, using the induction leads to $q^* \in \mathcal{C}_m, \forall m \in M$ under the event $E_0$.

Now, we define $i_\epsilon = \lceil \log_2(2/\epsilon) \rceil$, $I = \{0, \cdots, i_\epsilon\}$, and write $\epsilon_i = 2^{-i}, \forall i \in I$. Additionally, we use $\lceil x \rceil_2 = 2^{\lceil \log_2(x) \rceil}$ to denote a function that represents the smallest power of 2 greater than or equal to $x$. In the following, we define $m'_i, \forall i \in I \backslash \{0\}$,

$$m'_i = c \ \min \begin{cases} 4a\epsilon_i^{\rho-2} \left(d_{vc} \log\left(\theta_{q^*}\left(a\epsilon^\rho\right)\right) + \log\left(\frac{4\log_2(ca/\epsilon_i)}{\delta}\right)\right) \\ 4\left(\frac{\beta+\epsilon_i}{\epsilon_i^2}\right) \left(d_{vc} \log\left(\theta_{q^*}\left(\beta + \epsilon_i\right)\right) + \log\left(\frac{4\log_2(4c/\epsilon_i)}{\delta}\right)\right) \end{cases} \tag{27}$$

and $m_i = \lceil m'_i \rceil_2$. Moreover, we set $m_0 = 0$. Considering every $i \in I \backslash \{0\}$ with $m_i \in M$, combining equation 24, equation 25, $q^* \in \mathcal{C}_{m_i-1}$ and Line 9 in Algorithm 1, we obain the following results. Conditional on the event $E_0$, it holds that

$$\forall h \in V_{m_i}, er(h) - er(q^*) \leq 2\epsilon_i, \quad \forall i \in I \text{ with } m_i \in M. \tag{28}$$

Now, we turn to the analysis of the following label complexity

$$\sum_{m=1}^{\min\{m_{i_\epsilon}, \max M\}} \mathbb{1}_{\text{POS}(\mathcal{C}_{m-1}) \bigcup \text{DIS}(\mathcal{C}_{m-1})} (\mathbf{X}_m) = \sum_{i=1}^{i_\epsilon} \sum_{m=m_{i-1}+1}^{\min\{m_i, \max M\}} \mathbb{1}_{\text{POS}(\mathcal{C}_{m-1}) \bigcup \text{DIS}(\mathcal{C}_{m-1})} (\mathbf{X}_m)$$

$$\tag{29}$$

Furthermore, conditional on the event $E_0$, for each $i \in I \setminus \{0\}$ and $m \in \{m_{i-1}+1, \cdots, m_i\} \bigcap M$, we have DIS $(\mathcal{C}_{m-1}) \subseteq$ DIS $(\mathcal{C}_{m_{i-1}}) \subseteq$ DIS $(B(q^*, a(2\epsilon_{i-1})^\rho))$. The last subset inclusion results from Assumption D.1. Combined with the fact that POS $(\mathcal{C}_{m-1}) \subseteq \Omega_\gamma, \forall m \in [1, \min\{m_{i_\epsilon}, \max M\}]$, the summation of equation 29 is at most

$$\sum_{i=1}^{i_\epsilon} \sum_{m=m_{i-1}+1}^{m_i} \mathbb{1}_{\Omega_\gamma \bigcup \text{DIS}(B(q^*, a(2\epsilon_{i-1})^\rho))} (\mathbf{X}). \tag{30}$$

equation 30 represents the sum of independent Bernoulli *r.v.*. By using a Chernoff bound, the following event $E_1$ holds with probability at least $1 - \delta/3$,

$$\sum_{i=1}^{i_\epsilon} \sum_{m=m_{i-1}+1}^{m_i} \mathbb{1}_{\Omega_\gamma \bigcup \text{DIS}(B(q^*, a(2\epsilon_{i-1})^\rho))} (\mathbf{X})$$

$$\leq \log_2 (3/\delta) + 2e \sum_{i=1}^{i_\epsilon} (m_i - m_{i-1}) P \left( \Omega_\gamma \bigcup \text{DIS}\left(B\left(q^*, a\left(2\epsilon_{i-1}\right)^\rho\right)\right)\right)$$

$$\leq \underbrace{\log_2 (3/\delta) + 2e \sum_{i=1}^{i_\epsilon} (m_i - m_{i-1}) P\left(\Omega_\gamma\right)}_{\clubsuit} + \underbrace{2e \sum_{i=1}^{i_\epsilon} (m_i - m_{i-1}) P\left(\text{DIS}\left(B\left(q^*, a\left(2\epsilon_{i-1}\right)^\rho\right)\right)\right)}_{\spadesuit} \tag{31}$$

$\clubsuit$ in equation 31 characterizes the number of the unlabeled points sampled from $p_{\mathbf{X}}$ to achieve the excess error $\epsilon$ for a classifier returned by *MPED-RobustCAL*. Suppose the passive learning is used rather than querying POS $(\mathcal{C}) \bigcup$ DIS $(\mathcal{C})$ in *MPED-RobustCAL*. Then, the labels of all unlabeled points are queried, and $\clubsuit$ indicates the label complexity for passive learning with a constant factor $2eP(\Omega_\gamma)$ to achieve $\epsilon$. By using equation 21 and Definition 4.3, we have

$$\clubsuit \lessapprox 2eaP(\Omega_\gamma) \left(\frac{1}{\epsilon}\right)^{2-\rho} \left(d_{\text{vc}} \log\left(\theta_{q^*}(0)\right) + \log(1/\delta)\right) \tag{32}$$

In the theoretical analysis of original RobustCAL presented in Theorem 5.4 in Hanneke et al. (2014), $\log_2 (3/\delta) + \spadesuit$ in equation 31 represents the label complexity of the original RobustCAL. Furthermore, $P\left(\text{DIS}\left(B\left(q^*, a\left(2\epsilon_{i-1}\right)^\rho\right)\right)\right) \leq \theta_{q^*}(0) a(2\epsilon_{i-1})^\rho$ based on the Definition 4.3. Then, we restate their results in the following,

$$\log_2 (3/\delta) + \spadesuit \lessapprox \min \begin{cases} a^2 \theta_{q^*}(0) \epsilon^{2(\rho-1)} \left(d_{\text{vc}} \log\left(\theta_{q^*}(0)\right) + \log\left(\frac{\log(a/\epsilon)}{\delta}\right)\right) \log(1/\epsilon) \\ \theta_{q^*}(0) \left(\frac{\beta^2}{\epsilon^2} + \log\left(\frac{1}{\epsilon}\right)\right) \left(d_{\text{vc}} \log\left(\theta_{q^*}(0)\right) + \log\left(\frac{\log(1/\epsilon)}{\delta}\right)\right) \end{cases} \tag{33}$$

Combining equation 32 and equation 33, and plugging $\rho = 1$ given Assumption D.3 leads to the following label complexity

$$2eaP(\Omega_\gamma) \left(\frac{1}{\epsilon}\right)^{2-\rho} \left(d_{\text{vc}} \log\left(\theta_{q^*}(0)\right) + \log(1/\delta)\right) +$$

$$\min \begin{cases} a^2 \theta_{q^*}(0) \epsilon^{2(\rho-1)} \left(d_{\text{vc}} \log\left(\theta_{q^*}(0)\right) + \log\left(\frac{\log(a/\epsilon)}{\delta}\right)\right) \log(1/\epsilon) \\ \theta_{q^*}(0) \left(\frac{\beta^2}{\epsilon^2} + \log\left(\frac{1}{\epsilon}\right)\right) \left(d_{\text{vc}} \log\left(\theta_{q^*}(0)\right) + \log\left(\frac{\log(1/\epsilon)}{\delta}\right)\right) \end{cases} \tag{34}$$

equation 34 is expressed using big $\mathcal{O}$ notation in equation 4 in Theorem 4.5. By selecting the budget $B$ larger than equation 34, we ensure $m_{i_\epsilon} \in M$. Lastly, considering $P(E_0 \bigcap E_1) \geq P(E_0) + P(E_1) - 1 = 1 - \delta$, we have proved that for each $h$ in $\mathcal{C}$ returned by *MPED-RobustCAL*, $\text{er}(h) - \text{er}(q^*) \leq \epsilon$ with probability at least $1 - \delta$ using the label complexity in equation 34.

When $E_0 \bigcap E_1$ holds, the regions not included in POS $(\mathcal{C}) \bigcup$ DIS $(\mathcal{C})$ are those where points are classified as 0, given that $q^* \in \mathcal{C}_m, \forall m \in M$. Therefore, $\Omega_\gamma \subseteq$ POS $(\mathcal{C}) \bigcup$ DIS $(\mathcal{C})$. By the end of execution by *MPED-RobustCAL*, the excess error of any classifier in $\mathcal{C}$ returned by *MPED-RobustCAL* is upper-bounded by $\epsilon$ conditional on $E_0 \bigcap E_1$. Consequently, using the Definition 4.3, the ratio $\mathcal{R}$ of size of the enrollment region to size of $\Omega_\gamma$ is

$$\mathcal{R} = \frac{|\text{POS}(\mathcal{C}) \bigcup \text{DIS}(\mathcal{C})|}{|\Omega_\gamma|} \leq \frac{P(\Omega_\gamma) + P(\text{DIS}(\mathcal{C}))}{P(\Omega_\gamma)} \leq 1 + \frac{\theta_{q^*}\epsilon}{P(\Omega_\gamma)}. \tag{35}$$

This completes the proof. $\qquad \square$

## E  PROOF OF THEOREM 5.1

As we instantiate $k$ in *MPED-RobustCAL* with the sequential predictive two-sample test proposed in Podkopaev & Ramdas (2023), which is statistically valid under random enrollment (i.e., $\tilde{\mathbf{X}} \sim p_{\mathbf{X}}$), we aim to demonstrate that this statistical validity is preserved even when the test is conducted under *MPED-RobustCAL*. The same proof can be extended to any sequential test that is statistically valid under random enrollment.

*Proof.* The sequential predictive two-sample test, illustrated in Figure B.1, was first introduced in Podkopaev & Ramdas (2023). By combining Equations (5) and (11a) from that work, one can derive the test statistic defined in equation 5. The authors proved in Theorem 1 (first point) of Podkopaev & Ramdas (2023) that under the null hypothesis $H_0$—specifically, when the sample measurement $\mathbf{S}$ and group membership $A$ are independent (i.e., $\mathbf{S} \perp\!\!\!\perp A$)—the following bound holds:

$$P\left(\exists n \geq 1 : W_n \geq \frac{1}{\alpha}\right) \leq \alpha, \tag{36}$$

regardless of the choice of $\bar{q}$ used to construct the betting statistic $W_n$. To establish equation 36, it suffices to show that $(W)^n$ is a non-negative supermartingale under $H_0$.

**Definition E.1.** *(Supermartingale) A sequence $(Y)$ is a supermartingale if $\mathbb{E}\left[Y_{n+1} \mid Y^n\right] \geq Y_n, \forall n > 0$.*

Then, applying the Ville's inequality (Ville, 1939) immediately yields equation 36. We refer readers D.3 in Podkopaev & Ramdas (2023) for the same statement. Now, to reuse the result of equation 36 within *MPED-RobustCAL*, it remains to demonstrate that the sequence $(W)^n$ resulting from Algorithm 3 applied within *MPED-RobustCAL* is a non-negative supermartingale under $H_0$. As *MPED-RobustCAL* randomly assigns the treatment and control within $\left(\tilde{\mathbf{X}}, \tilde{\mathbf{X}}'\right)$, we have $\tilde{\mathbf{X}} \perp\!\!\!\perp A$. Additionally, under $H_0$ and Assumption 3.1, $\Delta(\mathbf{x}) = 0, \forall \mathbf{x} \in \mathcal{X}$, implying $Y^A(\mathbf{x}) = Y^{1-A}(\mathbf{x}), \forall \mathbf{x} \in \mathcal{X}$, which leads to $A \perp\!\!\!\perp Y^A(\tilde{\mathbf{x}})$. Consequently, $P\left(\tilde{\mathbf{O}}, A\right) = P\left(\tilde{\mathbf{X}}, Y^A, A\right) = P\left(\tilde{\mathbf{X}}, Y^A\left(\tilde{\mathbf{X}}\right)|A\right) P(A) = P\left(\tilde{\mathbf{X}}, Y^A\left(\tilde{\mathbf{X}}\right)\right) P(A) = P\left(\tilde{\mathbf{O}}\right) P(A)$. Therefore, $\tilde{\mathbf{O}} \perp\!\!\!\perp A$ holds within MPED-RobustCAL under $H_0$. In addition, as only one unit of $\left(\tilde{\mathbf{O}}, A\right)$ is included to $k$, we have $P(A_n = 1) = P(A_n = 0) = 0.5$. This leads to, for $\forall n > 0$ and $W_0 = 1$,

$$\mathbb{E}\left[W_n \mid (W)^{n-1}\right] = W_{n-1}\left(\left(1 + \lambda_n L_n\left(\tilde{\mathbf{O}}_n, 1\right)\right) P(A_n = 1) + \left(1 + \lambda_n L_n\left(\tilde{\mathbf{O}}_n, 0\right)\right) P(A_n = 0)\right)$$
$$= W_{n-1}. \tag{37}$$

Futheremore, it is easy to see $W_n \geq 0, \forall n > 0$ provided that $\lambda_n \in [-1, 1]$ and $L_n\left(\tilde{\mathbf{O}}, A\right) \in \{-1, 1\}$. Hence, under $H_0$, $(W)^n$ is a non-negative supermartingale regardless of the choice of $\bar{q}$ used to construct $L_n$, and this completes the proof. $\square$

