# OpenReview forum: "Matched-Pair Experimental Design with Active Learning"
_ICLR.cc/2026/Conference — Submitted to ICLR 2026_

### Official Review · Reviewer_eqTE · 2025-10-27

**Soundness:** 2
**Presentation:** 2
**Contribution:** 2
**Rating:** 4
**Confidence:** 3

**Summary:**

The paper introduces a framework called Matched-Pair Experimental Design with Active Learning (MPED-RobustCAL), which integrates agnostic active learning method RobustCAL into treatment-responder detection in matched-pair experimental designs. The motivation arises from the challenge that when the overall treatment effect in a population is small, conventional matched-pair designs may lack sufficient statistical power due to inefficient random sampling. The proposed approach reformulates the identification of high treatment-effect regions as a classification problem and employs an active learning strategy adapted from the RobustCAL algorithm to sequentially select participant pairs for experimentation. The authors provide theoretical guarantees showing that the enrollment region generated by MPED-RobustCAL encloses and converges to the true high-effect region, with improved label efficiency compared to passive learning, while maintaining statistical validity through Type I error control. Experiments conducted on synthetic data and two real-world datasets demonstrate that MPED-RobustCAL achieves higher testing power and true positive rates than conventional matched-pair designs and existing active designs such as regression-based active design and Bayesian active learning methods. The paper presents a theoretically supported and empirically validated method for improving the efficiency and effectiveness of matched-pair experimental designs to select regions with high treatment effects with limited experimental budgets.

**Strengths:**

The paper addresses the need of active learning in matched-pair experimental design, which is a reasonable concern. It introduces a framework that integrates Matched-Pair Experimental Design with active learning method RobustCAL to improve sampling efficiency. The theoretical analysis offers sounding guarantees on label efficiency, convergence, and statistical validity. The experimental results, conducted on both synthetic and real-world datasets, show consistent performance improvements in testing power and true positive rates.

**Weaknesses:**

- The novelty of the proposed method should be further clarified, as it primarily adapts existing active learning strategies to the matched-pair setting, with theoretical development largely following prior work on RobustCAL.

- Some assumptions, including those related to the data noise structure and the quality of matching, appear to be idealized and may not fully reflect realistic experimental conditions.

- Several parts of the paper, particularly the explanation of why the proposed method works, are not sufficiently clear for non-expert readers and would benefit from a more accessible presentation.

**Questions:**

- It is recommended that the authors provide a more detailed explanation of the novelty of the proposed method, particularly highlighting any key challenges required to apply RobustCAL to the Matched-Pair Experimental Design (MPED) setting.

- The paper appears to conflate individual outcomes with outcomes conditional on covariates. The hypothesis test in equation (2) seems to concern the equivalence of conditional outcome distributions $Y \mid a$ given treatment, which are not equivalent to the distributions of potential outcomes—a fundamental issue in causal inference. Since potential outcome distributions are generally unidentifiable, the paper would benefit from clearer terminology and a more precise explanation of what the key hypothesis test is evaluating.

- Assumption 4.4 is quite strong, as it may not hold in common cases where $\tilde{\eta}(\mathbf{X})$ is continuous in $\mathbf{X}$. It would be helpful if the authors could provide examples or scenarios where this assumption is valid.

- For Algorithm 1, clarification is needed on how the minimum in line 9 is computed. Would performing this search be computationally expensive in practice?

### Miscellaneous

- The phrase *treatment effect size* in line 96 is used ambiguously and may be misinterpreted as referring to the sample size. Replacing it with *value* or *magnitude of the treatment effect* is recommended.

- The notation $k_n$ in line 140 should likely be $k$.

- The term *corresponding outcomes* in line 196 is unclear. Do $(Y^0, Y^1)$ represent the potential outcomes of the unit with covariate $\tilde{\mathbf{X}}$ or of its matched pair $\tilde{\mathbf{X}}'$?

**Details Of Ethics Concerns:**

None.

---

> ### Author Response · Authors · 2025-11-25
>
> **On the novelty of the proposed design**\
> The practical novelty of our design lies in efficiently classifying the responding region while ensuring that most responders are identified—thus achieving high recall. This is clinically valuable because, when the responder region is small relative to the covariate space, conventional active designs tend to miss many responders, resulting in low recall. This motivation is stated in lines 67–70 and 73–77. This practical advantage is enabled by a key technical innovation: integrating the RobustCAL framework into our design by formulating the identification of the responding region as a classification problem. This is a substantial departure from existing regression-based approaches, which do not guarantee coverage of the responding region. We highlight this innovation in lines 61–64, and our recall results (true positive rate), such as those in Figure 3, further corroborate this advantage.
>
> **On the realism of the assumption 4.4**\
> There appears to be a misunderstanding of the broad applicability of Assumption 4.4. This assumption is classical and known as the Massart noise condition, which has been extensively used in the Probably Approximately Correct (PAC) learning framework. Examples include its use in Diakonikolas (2019) and the discussion in Section 2.3 of Hanneke (2014). The assumption is practical because it only requires the true classification probability $\tilde{\eta}(\mathbf{x})$ to be (slightly) bounded away from completely noisy level, ensuring that a detectable signal exists for all $\mathbf{x} \in \mathcal{X}$.
>
> Diakonikolas, Ilias, Themis Gouleakis, and Christos Tzamos. "Distribution-independent pac learning of halfspaces with massart noise." Advances in Neural Information Processing Systems 32 (2019).
>
> Steve Hanneke et al. Theory of disagreement-based active learning. Foundations and Trends® in Machine Learning, 7(2-3):131–309, 2014.
>
> **On the realism of high-matching quality**\
> First, many longitudinal studies obtain matched pairs by measuring responses for the same participant before and after treatment, which yields perfect matching. Our design remains broadly applicable in this setting, and our IHDP experiments fall into this category. Second, in practice, experimenters typically define baseline covariates such as gender, race, or other continuous variables and sample matched pairs conditional on these covariates ((see Bai, 2022, page 3912, second paragraph ). In this setting, even if the matching is imperfect, the labels are generated based on the baseline variables, and the label generation process for these variables is accurately captured. This leads to a classification problem defined on the baseline covariate space, and our theoretical results still apply. We have added this clarification (highlighted in red) in Section 4.1 of the manuscript.
>
> For what it's worth, assuming high-matching quality is standard in MPED and experimenters routinely apply data-quality checks to ensure sufficiently high matching accuracy and to mitigate the impact of mismatches. See (Simon, 2007) and (Grossarth-Maticek, 2006)
>
> Bai, Yuehao. "Optimality of matched-pair designs in randomized controlled trials." American Economic Review 112.12 (2022): 3911-3940.
>
> Simon, Laura J., and Vernon M. Chinchilli. "A matched crossover design for clinical trials." Contemporary clinical trials 28.5 (2007): 638-646.
>
> Grossarth-Maticek, R., and Renatus Ziegler. "Randomised and non-randomised prospective controlled cohort studies in matched-pair design for the long-term therapy of breast cancer patients with a mistletoe preparation (Iscador): a re-analysis." European journal of medical research 11.11 (2006): 485.
>
> **On the realism of other assumptions**\
> Assumption 3.1 specifies the absence or existence of a treatment effect under $H_0$ and $H_1$. Assumption 3.2 follows standard practice in causal inference to ensure unconfoundedness. We would be happy to provide further clarification if the reviewer could specify which aspects seem overly strong.

---

> ### Author Response · Authors · 2025-11-25
>
> **What our hypothesis testing is evaluating given the non-identifiability of potential outcomes**\
> This is an insightful point, and it reflects a gap in understanding the general purpose of our design within causal learning. Our design aims to detect the presence of a causal effect. The hypothesis concerns the equivalence of the marginal distributions of potential outcomes $Y^0$ and $Y^1$. In randomized controlled trials (RCT), unconfoundedness ensures that equivalence of observed treatment and control outcomes implies equivalence of the corresponding potential outcomes (see Duan, 2022). Conventional MPED improves efficiency by using matched treatment–control pairs to reduce the variance of observed $Y^1 - Y^0$. Our design further enhances efficiency by actively querying labels in local high–treatment-effect regions under $H_1$, improving the power of testing the nonequivalence of $p_{Y^0}$ and $p_{Y^1}$.
>
> Duan, Boyan, Aaditya Ramdas, and Larry Wasserman. "Interactive rank testing by betting." Conference on Causal Learning and Reasoning. PMLR, 2022.
>
> **On the computation of line 9 in Algorithm 1**\
> We provide a practical implementation in Algorithm 2 using committee-based active learning, which alleviates the computational cost of constructing a new $\mathcal{C}$ in line 9 of Algorithm 1. All empirical analyses are conducted using Algorithm 2. In our setting, label cost is the primary constraint, so computational efficiency is orthogonal to the study. The core computational step in line 9 of Algorithm 1 is evaluating $U(m, \delta)$, and Remark 4.2 clarifies that this quantity can be computed using Equation (20).
>
> **On the meaning of $(Y^0, Y^1)$**\
> Thank you for the great question. In MPED, we treat $\left(\tilde{\mathbf{X}}, \tilde{\mathbf{X}}'\right)$ as an individual block, and $\left(Y^0, Y^1\right)$ represent the potential outcomes for that block. For example, one randomly samples a covariate from $\left(\tilde{\mathbf{X}}, \tilde{\mathbf{X}}'\right)$, assigns the sampled covariate—whether $\tilde{\mathbf{X}}$ or $\tilde{\mathbf{X}}'$—to the control condition, and observes the corresponding $Y^0$. The counterfactual outcome for that same covariate is then the treatment response $Y^1$.

---

### Official Review · Reviewer_iumY · 2025-10-31

**Soundness:** 2
**Presentation:** 2
**Contribution:** 2
**Rating:** 4
**Confidence:** 3

**Summary:**

The paper presents MPED-RobustCAL, an active learning-based experimental design that demonstrates clear advantages over conventional passive learning and standard MPED approaches. Through extensive experiments on both synthetic and real-world datasets such as ALS and IHDP, the authors show that MPED-RobustCAL achieves higher testing power, improved precision, and faster convergence in identifying regions with high treatment effects. The method is robust across different classifiers and label budgets, and the theoretical analysis is thorough, with proofs supporting the main claims. The inclusion of sensitivity analyses and comparisons with other active learning baselines further strengthens the empirical evaluation.

However, the paper also has some limitations. The performance of MPED-RobustCAL, while generally superior, can be affected by label noise and imperfect ground-truth identification in real-world datasets, which may prevent the true positive rate from reaching its theoretical maximum. Additionally, the learning efficiency of MPED-RobustCAL is sometimes lower than that of the original RobustCAL due to extra label queries required for two-sample testing. The method’s reliance on certain hyperparameters and the need for careful tuning may also limit its practical applicability in some settings. Overall, while the approach is promising and well-supported, further work on reducing sensitivity to noise and hyperparameters would enhance its impact.

**Strengths:**

na

**Weaknesses:**

na

**Questions:**

na

---

> ### Author Response · Authors · 2025-11-25
>
> **On label noise and imperfect ground-truth identification**\
> Our active design is developed under label noise, as reflected in Assumption 4.4 (the bounded-noise condition).
>
> Regarding imperfect ground-truth identification:
> First, many longitudinal studies obtain matched pairs by measuring responses for the same participant before and after treatment, which yields perfect ground-truth identification. Our design remains broadly applicable in this setting, and our IHDP experiments fall into this category.
> Second, in practice, experimenters often define baseline covariates such as gender, race, or other continuous covariates, and they sample matched pairs conditional on these baseline variables (see Bai, 2022, page 3912, second paragraph). In this case, the labels or ground-truth contain no matching noise for the baseline variables. This leads to a classification problem defined on the baseline covariate space, and our theoretical results still apply. Therefore, assuming perfect label identification does not reduce the practical relevance of our approach.
>
> We have added this clarification (highlighted in red) in Section 4.1 of the manuscript.
>
> Bai, Yuehao. "Optimality of matched-pair designs in randomized controlled trials." American Economic Review 112.12 (2022): 3911-3940.
>
> **Lower-efficiency relative to RobustCAL**\
> We view the reduced efficiency relative to RobustCAL as a worthwhile trade-off for achieving higher testing power. This trade-off enables an innovative framework that simultaneously improves the testing power of the sequential two-sample test and efficiently identifies the enrollment region (compared with passive learning).
>
> **Reliance on hyper-parameters**\
> The three main hyper-parameters in our design are the classifier choice, the number of classifiers and the size of the data used for initialization. In contrast, the significance level $\alpha$ and the treatment-effect threshold $\gamma$ are determined by clinicians based on the controlled Type I error rate and the minimum clinically meaningful effect size.
>
> Sections B4.3, B4.4, and B5.3 report results with different classifier choices, and Section B.6 provides sensitivity analyses on the number of classifiers and the initialization size. Across all settings, our method consistently yields higher testing power and strong recall over various label budgets.

---

### Official Review · Reviewer_PCUW · 2025-10-31

**Soundness:** 3
**Presentation:** 3
**Contribution:** 2
**Rating:** 4
**Confidence:** 3

**Summary:**

The work proposes a new experimental setting to consider within the context of detecting the existence of a treatment effect. Additionally, the authors propose a new method based on an active learning design in order to perform this detection task.

**Strengths:**

**Clarity:** This work does a good job of providing the necessary background for the problem for readers with an ML background than a clinical background. Additionally, it seems that all notations and derivations are correct. The authors acknowledge the usage of LLMs for polishing the writing.

**Significance:**

1. This work is able to theoretically show that their method has greater label efficiency than passive learning techniques where the trial participants are randomly split into groups.

2. It provides insight into medically relevant problems and considerations that exist within the context of active experimental design. There are also preliminary experiments to validate the effectiveness of their method.

**Weaknesses:**

The main weaknesses in the paper stem from motivational and experimental rigor. The motivational issues translate to a seeming lack of novelty. This may partly be because of the introduction and related work being squeezed into < 1.5 pages.

**Novelty:**

I believe that the authors do not do a good job of distinguishing their contribution from prior work. For example, the authors mention the existence of a separate field known as the estimation of the conditional average treatment effect (CATE). The authors explicitly define CATE as quantifying the treatment effect size with a limited sample. The authors instead claim their work is more about detecting the existence of a treatment effect. Semantically, both lines of work can be considered as the same concept as the end result is understanding how a treatment impacts different subgroups of populations. I understand that CATE may be more of a post hoc analysis rather than the sequential online setting of this paper. However, if that is the case, then it should be possible to analyze how these CATE methods perform if molded into the framework of a sequential design which is not seen in the experimental setting.

The authors also base their method as an alteration on top of the Robust CAL active learning approach. Specifically, they identify that their method does an additional sampling step into the positive agreement region of a set of classifiers. This can be a significant contribution, but there is no explicit analysis that distinguishes their method from that of Robust CAL. Specifically, there is no intuition about the benefit of sampling from the positive agreement region and in fact they show that it results in lower label efficiency compared to the Robust CAL algorithm by itself. Their theoretical analysis compares against passive learning, but an in depth comparison against Robust CAL would do more to highlight their own novelty. This problem is further exacerbated by the lack of a comparison against Robust CAL in their empirical results section.

**Motivation:**

In general, I think the novelty issue is partially due to a lack of motivation regarding the significance of their method itself. For example, their opening figure shows that their method is enrolling targets from a superset of targets, but their isn’t an analysis or any intuition that makes it clear why sampling over this larger region is better. In fact, I wonder if this sampling strategy has potential downfalls for sampling outside the region where the treatment effect takes place.

There also isn’t specific intuition regarding the choice of adding onto the Robust CAL algorithm. There are many active learning frameworks such as BALD, CoreSet, entropy, etc. What is it about Robust CAL specifically that motivated its choice for this very specific medical context problem? The entire label complexity analysis is based on this paper, but I am still unsure why it was chosen.

**Experimental:**

The problems of Novelty and Motivation are also highlighted by the choice of experiments with respect to their method. The authors claim their method is an improvement on Robust CAL, but it doesn’t compare against it in any of the experiments. Instead it compares against regular MPED, a regression based active design strategy, and tau-BALD. This limits how much we can assess the novelty of the method in this paper, because it isn’t clear how the base strategy would have done. It would have also been natural to compare against standard active learning strategies such as Coreset, entropy, margin sampling, and other strategies that commonly appear in the literature. It isn’t clear to me why these methods could not have been adapted to this setting and given a fair basis of comparison.

I would have also been curious to establish differences amongst the selected targets that led to the improvement we observe empirically. For example, how did the samples with MPED-RobustCAL differ from those selected by MPED and can we visually or analytically show what covariates were sampled compared to those from the base algorithm.

I also think certain natural ablation studies are missing from this paper. For example, I understand that it is difficult to acquire data suitable for the MPED task, however, it would be good to perform a more comprehensive assessment with the synthetically generated data. For example, in the synthetic data, we could vary the treatment effect threshold and establish some intuition about how the method performs under different gamma values. Furthermore, it would have been good to establish how certain choices like s=0.5 corresponds to different real world clinical experimental conditions.

**Questions:**

1. Why wasn’t their a comparison with several active learning strategies? Is there something specific about BALD that admits its usage here, but not others?

2. Why wasn’t Robust CAL a comparison in the plots? I felt that would be a natural comparison.

3. Is there a way to intuitively demonstrate the advantages of sampling from both DIS and POS. I don’t fully understand why this implies enclosing the target region. I also want to see how this idea of DIS and POS evolves over training and potentially how this relates to certain distributions of covariates. Potentially, this would be a learning dynamics study of the 10 classifiers in the ensemble.

---

> ### Author Response · Authors · 2025-11-25
>
> Thank you for your insightful critique. We believe many concerns stem from a *misunderstanding of sequential design* in hypothesis testing. As noted in lines 134–140, a sequential two-sample test collects data sequentially and decides whether to reject $H_0$ (i.e., the absence of a treatment effect) *whenever new data arrive*. Under our setting of label scarcity, an intuitive active-sequential strategy is to query labels in high-effect regions so that, whenever a new label arrives, the sequential two-sample test is invoked using high–treatment-effect samples to determine whether to reject $H_0$, thereby improving testing power.
>
> The innovation of our active design is its unique ability to satisfy the following: (1) increase the testing power of the sequential test by sampling from the high-treatment effect region $\Omega_\gamma$ (2) efficiently identify $\Omega_\gamma$ (3) ensure high coverage of $\Omega_\gamma$. These have been explicitly stated in lines 72-80. We summarize the reviewer’s comments as the following questions, and provide responses that elaborate on the innovation.
>
> **On the lack of comparison with the CATE approaches**\
> We have indeed included comparisons with CATE approaches in our experiments. We respectfully remind the reviewer that one objective of our problem setup (see lines 188–189) is to define an enrollment strategy that sufficiently enrolls responders within a target region. This has substantial clinical value (see lines 67–70), because the true responding region is typically small and conventional designs miss many responders. An important outcome of our active design is an enrollment region that encloses this target region with high treatment effect. In Figure 3, our comparison with $\tau$-BALD and regression-based active design, which are two CATE approaches, shows consistently high recall during label querying, indicating that our design successfully includes most responders. In addition, the improved testing power in Table 1 compared with random sampling (conventional MPED) shows that the enrollment region effectively shrinks toward the region with high treatment effect.
>
> **On the lack of comparison with RobustCAL and the lower efficiency relative to RobustCAL**\
> RobustCAL, like standard active learning methods, does not improve the testing power of a sequential two-sample test. As clarified earlier, the sequential test evaluates whether to reject $H_0$ whenever new data arrive. Our design therefore queries samples in high treatment effect regions to increase the efficiency of the sequential test, which is a different objective from learning a classifier. The lower efficiency relative to RobustCAL is the necessary tradeoff for achieving higher testing power in the two-sample test.
>
> Consequently, our sampling region includes both $\text{POS}(\mathcal{C})$, which enhances testing power, and $\text{DIS}(\mathcal{C})$, which improves classifier learning. Typical active learning methods sample mainly from $\text{DIS}(\mathcal{C)}$, a classification uncertainty region that does not necessarily contain high treatment effect samples, and thus cannot effectively increase testing power, as shown in prior work (Li 2022, Figure 2).
>
> In summary, our active design uniquely improves two forms of efficiency: the testing power of the sequential two-sample test and the classifier-learning efficiency relative to random sampling.
>
> Li, Weizhi, et al. "A label efficient two-sample test." Uncertainty in Artificial Intelligence. PMLR, 2022.
>
> **On the motivations of enrolling from the superset, $\text{POS}(\mathcal{C})\bigcup \text{DIS}(\mathcal{C})$, of target regions**\
> The reviewer’s concern about sampling from the superset $\text{POS}(\mathcal{C})\bigcup \text{DIS}(\mathcal{C})$ arises from a *misunderstanding* of the objectives stated in our problem setup (lines 185–189). Under $H_1$, our objectives are twofold: (1) to improve the testing power of the sequential two-sample test, and (2) to accurately identify the high-treatment-effect region $\Omega_\gamma$. Our active design is the only approach that simultaneously achieves both goals by:
> 1. Improving testing power  through querying labels in $\text{POS}(\mathcal{C})$, which approximates the high-treatment-effect region $\Omega_\gamma$; and
> 2. Facilitating accurate classification of $\Omega_\gamma$ by also sampling in $\text{DIS}(\mathcal{C)}$. This step is aligned with what conventional active learning algorithms typically do.
>
> Sampling exclusively in $\text{POS}(\mathcal{C})$ or exclusively in $\text{DIS}(\mathcal{C})$ would fail to achieve one of these two objectives, whereas sampling from their union enables both.

---

> ### Author Response · Authors · 2025-11-25
>
> **On the Motivation for choosing RobustCAL and why not other active learning (AL) algorithms**\
> This is an important point to clarify. RobustCAL is arguably one of the earliest and most formal frameworks for analyzing the theoretical label complexity of agnostic AL. Many existing AL algorithms including committee-based and entropy-based approaches can be viewed as practical instantiations of the RobustCAL framework (See Section 5.3 of Hanneke 2014). BALD, in contrast, is a Bayesian AL method and is therefore conceptually distinct from the frequentist-based RobustCAL. A key advantage of RobustCAL for our problem is that the disagreement region $\text{DIS}(\mathcal{C})$ contains a positive region that is not captured by $\text{POS}(\mathcal{C})$. As a result, $\text{DIS}(\mathcal{C}) \bigcup \text{POS}(\mathcal{C})$ forms an enclosing region of the target high-treatment-effect region $\Omega_\gamma$. This property is essential to our design: it ensures that all positive examples (i.e., responding patients) are included in the enrollment region, leading to high recall in the classification task. Such coverage is not guaranteed by other standard AL algorithms. Theorem 4.5 formalizes this coverage property, which follows from the RobustCAL guarantee that an optimal classifier is always included in $\mathcal{C}$ (Lemma D.5; see also Section 5.2 of Hanneke, 2014).
>
> Steve Hanneke et al. Theory of disagreement-based active learning. Foundations and Trends® in Machine Learning, 7(2-3):131–309, 2014.
>
> **On the missing ablation studies on $s$ and $\gamma$ in the synthetic dataset**\
> We have included an ablation study of $s$ in Table 8, showing that when $s$ varies from 0.1 to 0.5, our proposed design consistently maintains a testing-power gain over the conventional MPED under the same label budget. We also provide supplementary results below for $\gamma$ values from 0.3 to 0.8, demonstrating both higher testing power and higher recall compared with the baselines. The choice of $\gamma$ depends on the clinically meaningful effect size in practice; larger values yield greater testing-power gains by concentrating enrollment in higher treatment-effect regions.
>
> | Method             | $\gamma$ = 0.3 | $\gamma$ = 0.4 | $\gamma$ = 0.5 | $\gamma$ = 0.6 | $\gamma$ = 0.7 | $\gamma$ = 0.8 |
> |--------------------|----------------|----------------|----------------|----------------|----------------|----------------|
> | Conventional MPED  | 0.07           | 0.07           | 0.07           | 0.07           | 0.07           | 0.07           |
> | MPED-RobustCAL     | **0.18**       | **0.29**       | **0.46**       | **0.57**       | **0.71**       | **0.79**       |
> **Table:** Testing-power comparison between Conventional MPED and MPED-RobustCAL at label budget 200 across $\gamma$, averaged over 100 runs. *Note: Conventional MPED has constant testing power due to its random enrollment strategy.*
>
> | Method                         | $\gamma$ = 0.3 | $\gamma$ = 0.4 | $\gamma$ = 0.5 | $\gamma$ = 0.6 | $\gamma$ = 0.7 | $\gamma$ = 0.8 |
> |--------------------------------|----------------|----------------|----------------|----------------|----------------|----------------|
> | Regression-based Active Design | 0.7714         | 0.5952         | 0.3402         | 0.1507         | 0.0618         | 0.0193         |
> | $\tau$-BALD                    | 0.7640         | 0.5864         | 0.3684         | 0.1771         | 0.0846         | 0.0201         |
> | MPED-RobustCAL                 | **0.9705**     | **0.9981**     | **0.9930**     | **0.9897**     | **0.9812**     | **0.9845**     |
> **Table:** True positive rate (recall) comparison between the proposed MPED-RobustCAL and two other active designs across $\gamma$ at label budget = 50, averaged over 100 runs.
>
> **On the advantages of sampling from $\text{DIS}(\mathcal{C}) \bigcup \text{POS}(\mathcal{C})$ and its progression**\
> As noted in our response to Q3, sampling from $\text{DIS}(\mathcal{C}) \bigcup \text{POS}(\mathcal{C})$ jointly ensures high testing power and efficient classifier learning. Moreover, as shown in Figures 3, 9, and 10,  $\text{DIS}(\mathcal{C}) \bigcup \text{POS}(\mathcal{C})$ maintains high recall throughout the label-query process, indicating that a large proportion of target examples are consistently captured. Figure 5(b) visualizes the distribution of the queried samples, showing that the enrollment region produced by the proposed approach more tightly encloses the target region (in red) than the conventional approach, i.e., random sampling.

---

### Official Review · Reviewer_aG3c · 2025-11-01

**Soundness:** 3
**Presentation:** 3
**Contribution:** 3
**Rating:** 4
**Confidence:** 3

**Summary:**

The paper proposes MPED-RobustCAL, an active-learning framework for matched-pair experimental designs (MPEDs) aimed at detecting (rather than estimating) treatment effects when the overall ATE is small but there exists a responsive subregion Ω_γ. The key idea is to recast “find the responders” as a binary classification task.

- Kato, Oga, Komatsubara Inokuchi "Active adaptive experimental design for treatment effect estimation with covariate choice." ICML2024

**Strengths:**

This study proposes an innovative approach to detecting treatment effects in matched-pair experimental designs (MPEDs) and provides sufficient theoretical results, supported by empirical analyses.

**Weaknesses:**

I am interested in the relationship between treatment-effect estimation and treatment-effect detection. For example, Kato et al. (2024) present an active, adaptive experimental design for efficient estimation of the average treatment effect. Although the experimental settings differ, the two lines of work are closely related in that both study active, sequential designs for treatment effects. Can the method proposed here be applied to that setting, and what is the precise relationship between them?

**Questions:**

See Weakness section.

---

> ### Author Response · Authors · 2025-11-25
>
> **On the connection between our active design for effect detection and active designs for treatment-effect estimation (e.g., Kato 2024).**\
> Thank you for pointing out this important line of work. Our design, however, is not applicable to the treatment-effect estimation setting of Kato (2024). Effect detection differs fundamentally from estimation: detection only requires determining whether a local effect exists, while estimation requires accurately quantifying its magnitude. Active detection methods typically enroll units from high-effect regions, which introduces selection bias and leads to a biased estimator, making such strategies unsuitable for active estimation. A *key goal* of  our setting (see lines 188-189) is to develop an enrollment scheme that captures all treatment-responding patients. Our design not only improves detection efficiency but also ensures that the enrollment region encloses and efficiently converges to the true responding region. This yields high recall even when the responding region is small (see lines 67–71). To the best of our knowledge, our design is the *only active design* that achieves both efficiency gains and full coverage of the responding region.

---

### Meta-Review · Area_Chair_SvDM · 2026-01-06

**Summary:**

Reviewer aG3c wanted to see a discussion on the connection between treatment effect estimation and treatment effect detection.

Reviewer PCUW had several concerns: RobustCAL comparison is missing from both theoretical and experimental results. CATE comparison is missing/unclear. Ablation studies are missing.

Reviewer iumY had a similar concern about performance relative to RobustCAL and raised the issue of hyperparameters.

Reviewer eqTE did not mention CATE but raised a similar point about the authors confusing conditional treatment effect with the unconditional one.

Overall I think a lack of rigorous comparison with CATE estimation and how the proposed classification based framework is more adventageous than the large body of work on CATE estimation in this active setting is concerning.

**Reviewer Concerns:**

Reviewer PCUW's concerns around baselines and CATE comparison are not fully addressed.

Reviewer iumY may have been satisfied with their hyperparameter comment since authors pointed to result in appendix of the paper.

**Reviewer Scores:**

Reviewer PCUW may have been somewhat satisfied with the ablation studies but I dont' think they would have been satisfied with the authors' insistence on not comparing with RobustCAL. I understand authors are making a desired tradeoff but they can still compare to empirically showcase that tradeoff.

---

### Decision · Program_Chairs · 2026-01-26

Reject